# LASSIE: Learning Articulated Shapes from Sparse Image Ensemble via 3D Part Discovery

Chun-Han Yao[1]    Wei-Chih Hung[2]    Yuanzhen Li[3]    Michael Rubinstein[3]
Ming-Hsuan Yang[134]    Varun Jampani[3]

[1]UC Merced    [2]Waymo    [3]Google Research    [4]Yonsei University

## Abstract

Creating high-quality articulated 3D models of animals is challenging either via manual creation or using 3D scanning tools. Therefore, techniques to reconstruct articulated 3D objects from 2D images are crucial and highly useful. In this work, we propose a practical problem setting to estimate 3D pose and shape of animals given only a few (10-30) in-the-wild images of a particular animal species (say, horse). Contrary to existing works that rely on pre-defined template shapes, we do not assume any form of 2D or 3D ground-truth annotations, nor do we leverage any multi-view or temporal information. Moreover, each input image ensemble can contain animal instances with varying poses, backgrounds, illuminations, and textures. Our key insight is that 3D parts have much simpler shape compared to the overall animal and that they are robust w.r.t. animal pose articulations. Following these insights, we propose LASSIE, a novel optimization framework which discovers 3D parts in a self-supervised manner with minimal user intervention. A key driving force behind LASSIE is the enforcing of 2D-3D part consistency using self-supervisory deep features. Experiments on Pascal-Part and self-collected in-the-wild animal datasets demonstrate considerably better 3D reconstructions as well as both 2D and 3D part discovery compared to prior arts. Project page: https://chhankyao.github.io/lassie/

## 1   Introduction

The last decade has witnessed significant advances in estimating human body pose and shape from images [26, 2, 15, 19, 18, 34, 17]. Much progress is fueled by the availability of rich datasets [14, 29, 39] with 3D human scans in a variety of shapes and poses. In contrast, 3D scanning wild animals is quite challenging, and existing animal datasets such as SMALR [48] rely on man-made animal shapes. Manually creating realistic animal models for a wide diversity of animals (e.g. mammals and birds) is also challenging and time-consuming. In this work, we propose to automatically estimate 3D shape and articulation (pose) of animals from only a sparse set of image ensemble (collection) without using any 2D or 3D ground-truth annotations. Our problem setting is highly practical as each image ensemble consists of only a few (10-30) in-the-wild images of a specific animal species. Fig. 1 shows sample zebra images which forms our input. In addition, we assume a human-specified 3D skeleton (left in Fig. 1) that can be quite rough with incorrect bone lengths and this mainly provides the connectivity of animal parts. For instance, we use the same 3D skeleton for all the quadruped animals despite considerable shape variations across species, e.g., giraffe vs. zebra vs. elephant. Manually specifying a rough skeleton is a trivial task which only takes a few minutes.

Estimating articulated 3D shapes from in-the-wild image ensemble is a highly ambiguous and challenging problem. There are several varying factors across images: backgrounds; lighting; camera

36th Conference on Neural Information Processing Systems (NeurIPS 2022).

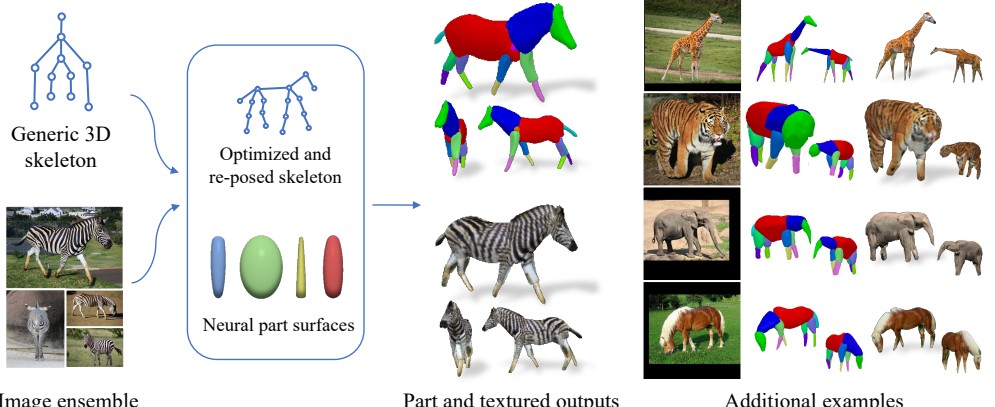

Generic 3D skeleton

Optimized and re-posed skeleton

Neural part surfaces

Image ensemble

Part and textured outputs

Additional examples

Figure 1: **Articulated shape optimization from sparse images in-the-wild.** Given 10-30 images of an articulated class and a generic 3D skeleton, we optimize the shared skeleton and neural parts as well as the instance-specific camera viewpoint and bone transformations. Our method is able to produce high-quality outputs without any pre-defined shape model or instance-specific annotations. The part-based representation also allows applications like texture and pose transfer, animation, etc.

viewpoints; animal shape, pose and texture, etc. In addition, our highly practical problem setting does not allow access to any 3D animal models, per-image annotations such as keypoints and silhouettes, or multi-view images.

In this work, we present LASSIE, an optimization technique to **L**earn **A**rticulated **S**hapes from **S**parse **I**mage **E**nsemble. Our key observation is that shapes are composed of 3D parts with the following characteristics: 1) 3D parts are geometrically and semantically consistent across different instances. 2) Shapes of 3D parts remain relatively constant w.r.t. changes in animal poses. 3) Despite the complex shape of overall animal body, 3D parts are usually made of simple convex shapes. Following these observations, LASSIE optimizes 3D parts and their articulation instead of directly optimizing the overall animal shape. Specifically, LASSIE optimizes a 3D skeleton and part shapes that are shared across instances; while also optimizing instance-specific camera viewpoint, pose articulation, and surface texture. To enforce semantic part consistency across different images, we leverage deep features from a vision transformer [10] (DINO-ViT [6]) trained in a self-supervised fashion. Recent methods [1, 38] demonstrate good 2D feature correspondences in DINO-ViT features and we make use of these correspondences for 3D part discovery. To regularize part shapes, we constrain the optimized parts to a latent shape manifold that is learned using 3D geometric primitives (spheres, cylinders, cones etc.). Being self-supervised at the image level, LASSIE can easily generalize to a wide range of animal species with minimal user intervention.

We conduct extensive experiments on image ensembles from the Pascal-Part Dataset [7] with quantitative evaluations using 2D ground-truth part segmentation and keypoints. We also collect in-the-wild web image ensembles with more animal species and annotate 2D keypoints for evaluation. Both qualitative and quantitative results demonstrate considerably better 3D reconstructions compared to the prior arts that use even more supervision. In addition, estimation of explicit 3D parts enables easy editing and manipulation of animal parts. Fig. 1 shows sample 3D reconstructions with discovered parts from LASSIE. The main contributions of this work are:

- To the best of our knowledge, this is the first work that recovers 3D articulated shapes from in-the-wild image ensembles without using any pre-defined animal shape models or instance-specific annotations like keypoints or silhouettes.
- We propose a novel optimization framework called LASSIE, where we incorporate several useful properties of mid-level 3D part representation. LASSIE is category-agnostic and can easily generalize to a wide range of animal species.
- LASSIE can produce high-quality 3D articulated shapes with state-of-the-art results on existing benchmarks as well as self-collected in-the-wild web image ensembles.

## 2   Related Work

**Animal shape and pose estimation.** Estimating 3D pose and shape of animal bodies from in-the-wild images is highly ill-posed and challenging due to large variations across different classes, instances,

viewpoints, articulations, etc. Most existing mesh reconstruction work [21, 24, 11, 36, 44] focus more on compact shapes like birds and cars, and thus cannot handle articulations of animal bodies. Recent articulation-aware methods, on the other hand, deal with a simplified scenario by assuming a pre-defined class-level statistical model, instance-specific images, or accessible human annotations. For instance, Zuffi *et al.* [49] build a statistical shape model, SMAL, for common quadrupedal animals, similar to the SMPL [26] model for human bodies. Follow-up work either optimize the SMAL parameters based on human annotated images [48] or train a neural network on large-scale image datasets to directly regress the parameters [47, 35]. However, the output shapes are limited by the SMAL shape space or training images which contain one or few animal classes. Kulkarni *et al.* [20] propose A-CSM technique that align a known template mesh with skinning weights to input 2D images. Other recent approaches [41, 40, 42] exploit the dense temporal correspondence in video frames to reconstruct articulated shapes. In contrast to existing methods, we deal with a novel and practical problem setting: learn without any pre-defined shape model, instance-level human annotations, or temporal information to leverage.

**Part discovery from image collections.** Deep feature factorization (DFF) [9] shows that one could automatically obtain consistent part segments by clustering intermediate deep features of a classification network trained on ImageNet. Inspired by DFF, SCOPS [13] learns a 2D part segmentation network from image collections, which can be used as semantic supervision for 3D reconstruction [24]. Choudury *et al.* [8] proposed to use contrastive learning for 2D part discovery. Lathuilière *et al.* [22] exploit motion cues in videos for part discovery. Recently, Amir *et al.*[1] demonstrate that clustering self-supervisedly learned vision transformers (ViT) such as DINO [6] features can provide good 2D part segmentations. In this work, we make use of DINO features for 3D part discovery instead of 2D segmentation. In the 3D domain, Tulsiani *et al.* [37] use volumetric cuboids as part abstractions to learn 3D reconstruction. Mandikal *et al.* [28] predict part-segmented 3D reconstructions from a single image. In addition, Luo *et al.* [27] and Paschalidou *et al.* [31, 32] learn to form object parts by clustering 3D points. Considering that most 3D approaches require ground-truth 3D shape of a whole object or its parts as supervision, Yao *et al.* [43] propose to discover and reconstruct 3D parts automatically by learning a primitive shape prior. Most of these works assume some form of supervision in terms of 3D shape or camera viewpoint. In this paper, we aim to discover 3D parts of articulated animals from image ensembles without any 2D or 3D annotations, which, to the best of our knowledge, is unexplored in the literature.

**3D reconstruction from sparse images.** Optimizing a 3D scene or object from multi-view images is one of the widely-studied problems in computer vision. The majority of recent breakthroughs are based on the powerful Neural Radiance Field (NeRF) [30] representation. Given a set of multi-view images, NeRF can learn a neural volume from which one can render high-quality novel views. Closely related to our work, NeRS [45] optimizes a neural surface representation from a sparse set of image collections and also only requires rough camera initialization. In this work, we use a neural surface representation to represent animal parts. Existing methods assume multi-view images captured in the same illumination setting and also the object appearance (texture) to be same across images. Another set of work [3, 46, 5] propose neural reflectance decomposition on image collections captured in varying illuminations. But they work on rigid objects with ground-truth segmentation and known camera poses. A concurrent work, SAMURAI [4] jointly reasons about camera pose along with shape and materials of rigid objects from image collections. In contrast, the input for our method consists of in-the-wild animal images with varying textures, viewpoints and pose articulations and captured in different environments.

## 3 Approach

Given a generic 3D skeleton and few images of an articulated animal species, LASSIE optimizes the camera viewpoint and articulation for each instance as well as the resting canonical skeleton and part shapes that are shared across all instances. We introduce our neural part representation in Section 3.1 and the optimization framework in Section 3.2. An overview of LASSIE is illustrated in Fig. 2.

### 3.1 Articulated Shapes with Neural Part Surfaces

**3D skeleton.** Constructing a template mesh or statistical shape model for specific animal classes requires either the ground-truth 3D shapes or extensive human annotation, which are both hard to obtain in real-world scenarios. A 3D skeleton, on the other hand, is easy to annotate since it

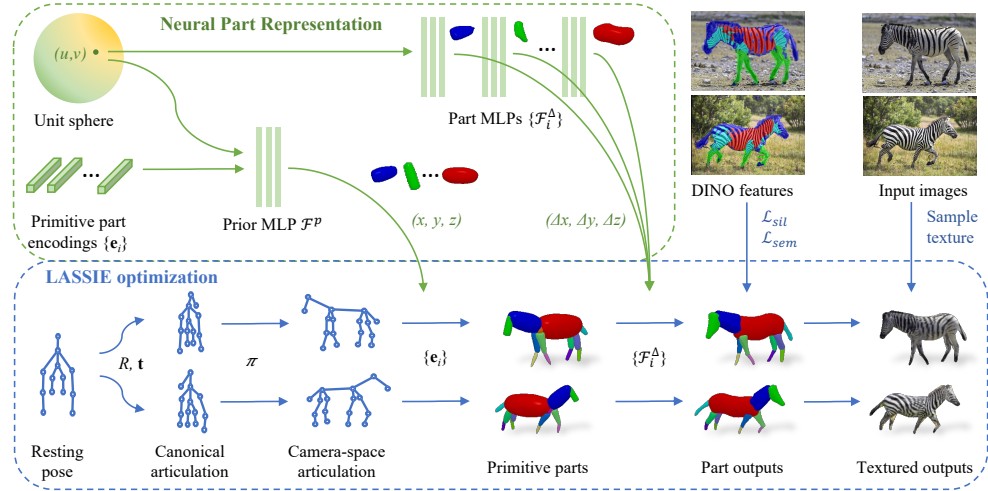

Figure 2: **Neural part surface representation.** Based on an optimized 3D skeleton, we reconstruct the articulated shape by optimizing the primitive latent codes and part deformation decoders. The final output is the composition of neural part surfaces with textures sampled from the input images.

only contains a set of sparse joints connected by bones. Furthermore, it generalizes well across classes, and one can effortlessly modify certain joints or bones to adapt to distinct animal types (e.g., mammals vs. birds). Assuming a simple and generic 3D skeleton is given, we propose a part-based representation for articulated shapes. A skeleton can be defined by a set of 3D joints $P \in \mathbb{R}^{p \times 3}$ and bones $B = \{(P_i, P_j)\}^b$ in a tree structure, where $p$ and $b = p - 1$ are numbers of joints and bones/parts, respectively. For instance, we use the same base skeleton for all four-legged animals in our experiments. The base skeleton specifies the connection of 16 joints and the initial joint coordinates in the root-relative 3D space. The skeleton is shared by all instances in the same class and can be optimized by scaling the bone lengths and rotating the bones with respect to the parent joints. Unlike the linear or neural blend skinning models [40, 42] where bones are unconstrained Gaussian components, our skeleton-based representation allows us to regularize bone transformations, discover meaningful and connected parts, and produce realistic shape animations.

**Neural part surfaces.** Given a skeleton which specifies the 3D joints and bones, we represent the articulated shape as a composition of several neural parts. Specifically, each part is defined as a neural surface wrapped around a skeleton bone. Motivated by object representation in NeRS [45], we represent part surface as a multi-layer perceptron (MLP) network that predicts the 3D deformation of a given continuous UV coordinate on the unit sphere. A key advantage of modeling parts instead of whole animal is that part shapes are usually of simple convex shapes and can be represented well with a deformed sphere. Compared to explicit mesh representation, the neural representation enables efficient mesh regularization on the output shapes while producing high resolution surfaces as MLPs can work with meshes of arbitrary vertex resolution. Each part surface is first reconstructed in the canonical space, then scaled by the corresponding bone length, rotated by the bone orientation, and centered at the bone centroid. This ensures that the parts are connected under various bone transformations and also easy to repose. Formally, let $X \in \mathbb{R}^{3 \times m}$ denote the $m$ uniformly distributed 3D vertices on a unit sphere. Further let $R_i \in \mathbb{R}^{3 \times 3}$, $\mathbf{t}_i \in \mathbb{R}^3$ and $s_i \in \mathbb{R}$ denote the global rotation, translation and scaling of the $i^{th}$ part ($i \in \{0, \cdots, b\}$), where $b$ denotes the number of parts. We first deform the sphere to obtain part shape using the part MLP, $X \rightarrow \mathcal{F}_i(X)$; followed by global transformation to obtain part vertices $V_i \in \mathbb{R}^{3 \times m}$ in the global coordinate frame:

$$V_i = s_i R_i \mathcal{F}_i(X) + \mathbf{t}_i. \tag{1}$$

**Latent part prior.** Although fitting neural part surfaces on sparse images can produce reasonable outputs that are faithful to the input view, we observe that part shapes tend to be non-uniform or unrealistic from other novel views. To further regularize the part shapes, we decompose the part MLP $\mathcal{F}_i$ into a prior MLP, $\mathcal{F}^p$ and a part deformation MLP, $\mathcal{F}_i^\Delta$. The prior MLP is trained to produce the base shapes that are close to geometric primitives such as spheres, cylinders, etc. The deformation MLP is then used to provide additional deformations on top of the base shapes. Fig. 2 illustrates these

two MLPs. As shown in the figure, the prior MLP takes as input the surface coordinate as well as a $d$-dimensional latent shape code $\mathbf{e}_i \in \mathbb{R}^d$. Both the latent space and the prior MLP are trained using geometric primitive shapes. Concretely, we train a Variational Auto-Encoder (VAE) which takes a primitive mesh as input, predicts the latent shape encodings and reconstructs the primitive through a conditional neural surface decoder which forms our prior MLP. The primitive meshes in this work are randomly sampled geometric primitives (spheres, ellipsoids, cylinders, and cones). With the use of prior and deformation MLPs, Eq. 1 now becomes:

$$V_i = s_i R_i(\mathcal{F}^p(X, \mathbf{e}_i) + \mathcal{F}_i^\Delta(X)) + \mathbf{t}_i, \tag{2}$$

where $\mathbf{e}_i$ denotes the latent shape encodings of part $i$.

## 3.2 Discovering 3D Neural Parts from Image Ensemble

**Optimization setting.** The input to our optimization is a sparse set (typically 20-30) of $n$ in-the-wild images $\{I_j\}_{j=1}^n$ along with a generic/rough 3D skeleton $P \in \mathbb{R}^{p \times 3}$ that has $p$ joints and $b$ bones/parts. The only image-level annotations for optimization are from the self-supervisory DINO [6] features. All the instances in the input image ensemble are of same species, but could have varying appearance due to texture, pose, camera angle, lighting, etc. We also assume that the entire animal body is visible in each image without truncated or occluded by another object although some degree of self-occlusion is allowed. Let $j$ denote the index over images $j \in \{1, ., n\}$ and $i$ denote the index over parts $i \in \{1, ., b\}$. For each image, LASSIE optimizes the camera viewpoint $\pi^j = (R_0, t_0)$ and part rotations $R^j \in \mathbb{R}^{b \times 3 \times 3}$. In addition, LASSIE optimizes these variables for each of the $i^{th}$ part that are shared across all the animal instances: bone length scaling $s_i \in \mathbb{R}$, latent shape encodings $\mathbf{e}_i \in \mathbb{R}^d$ and part deformation MLPs $\mathcal{F}_i^\Delta$. That is, we assume that part shapes are shared across all instances whereas their articulation (pose transformation) is instance dependent. Although this may not be strictly valid in practice, this provides an important constraint for our highly ill-posed optimization problem.

**Analysis by synthesis.** Since we do not have access to any form of 3D supervision, we use analysis-by-synthesis to drive the optimization. That is, we use differentiable rendering to render the optimized 3D articulated shape and define loss functions w.r.t. input images. Using Eq. 2, we first obtain the complete articulated shape for each image as a combination of different part meshes. We then use a differentiable mesh renderer [25] to project the 3D points onto 2D image space via the learnable camera viewpoints. A key challenge is that we do not have access to any from of 2D annotations such as keypoints or part segmentation.

**Self-supervisory deep features to the rescue.** Since different instances in the ensemble can have different texture, we do not model and render image pixel values. Instead, we rely on learned deep features for our self-supervisory analysis-by-synthesis framework. Recent works [1, 12] demonstrate that deep features from DINO [6] vision transformer are semantically descriptive, robust to appearance variations, and higher-resolution compared to CNN features [1, 38]. In this work, we use the *key* features from the last layer of the DINO network as semantic visual features for each image. We assume that these DINO features are relatively similar for same parts across different instances. In addition to using DINO features for consistent semantic features across images, we also compute 2D parts and foreground animal mask following similar steps as in [1]. Concretely, we take the class tokens from the last layer, compute the mean attention of class tokens as saliency maps, and sample the keys of image patches with high saliency scores. We then collect the salient keys from all images and apply an off-the-shelf K-means clustering algorithm to form $c$ semantic part clusters. Finally, we obtain foreground silhouettes by simply thresholding the pixels with minimum distance to cluster centroids. Fig. 3 shows the DINO part clusters on sample images from different image ensembles. We observe that using 4 clusters works well in practice. With the use of self-supervised features (keys) and rough silhouettes, we alleviate the dependency of ground-truth segmentation mask and keypoint annotations in our optimization framework.

**Silhouette loss.** We compute the silhouette loss $\mathcal{L}_{mask}$ by rendering all part surfaces with a differentiable renderer [25] and comparing with the pseudo ground-truth foreground masks obtained via clustering DINO features: $\mathcal{L}_{mask} = \sum_j \|M^j - \hat{M}^j\|^2$, where $M^j$ and $\hat{M}^j$ are the rendered and pseudo ground-truth silhouettes of instance $j$, respectively. Although these silhouette-based losses encourage the overall shape to match the 2D silhouettes, their supervisory signal is often too coarse to resolve the ambiguity caused by different camera viewpoints and pose articulations. Therefore, we design a novel semantic consistency loss to more densely supervise the reconstruction.

**2D-3D semantic consistency.** We propose a novel strategy to make use of the semantically consistent 2D DINO features for 3D part discovery. At a high-level, we impose the 2D feature consistency across the images via learned 3D parts. For this, we propose an Expectation-Maximization (EM) like optimization strategy where we alternatively estimate the 3D part features (E-step) from image ensemble features and then use these estimated 3D part features to update the parts (M-step). More formally, let $K^j \in \mathbb{R}^{h \times w \times f}$ denote the $f$-dimensional DINO key features from $j^{th}$ image and $Q \in \mathbb{R}^{m \times f}$ denote the corresponding features on the 3D shape with total $m$ vertices. In the E-step, we update $Q$ by projecting the 3D coordinates onto all the 2D image spaces via differentiable rendering under the current estimated shape and camera viewpoints. We aggregate the 2D image features from the image ensemble corresponding to each 3D vertex to estimate $Q$. In the M-step, we optimize the 3D parts and camera viewpoints using a 2D-3D semantic consistency loss $\mathcal{L}_{sem}$. To define $\mathcal{L}_{sem}$, we discretize the foreground 2D coordinates in each image $\{p | \hat{M}^j(p) = 1\}$. Likewise, we sample a set of points $\{v \in V^j\}$ on the 3D part surfaces and obtain their 2D projected coordinates $\pi^j(v)$. $\mathcal{L}_{sem}$ is then defined as the Chamfer distance in a high-dimensional space:

$$\mathcal{L}_{sem} = \sum_j \Big( \sum_{p | \hat{M}^j(p) = 1} \min_{v \in V^j} \mathcal{D}(p, v) + \sum_{v \in V^j} \min_{p | \hat{M}^j(p) = 1} \mathcal{D}(p, v) \Big). \tag{3}$$

For each instance $j$, the distance $\mathcal{D}$ between an image point $p$ and 3D surface point $v$ is defined as:

$$\mathcal{D}(p, v) = \underbrace{\|\pi^j(v) - p\|^2}_{\text{Geometric distance}} + \alpha \underbrace{\|Q(v) - K^j(p)\|^2}_{\text{Semantic distance}}, \tag{4}$$

where $\alpha$ is a scalar weighting for semantic distance. In effect, $\mathcal{L}_{sem}$ optimizes the 3D part shapes such that the aggregated 3D point features in the E-step would project closer to the similar pixel features in the image ensemble. We alternate between E and M steps every other iteration in the optimization process. In Eq. 4, the image coordinates $p$ and semantic features $K, Q$ are fixed in each optimization iteration, hence $\mathcal{L}_{sem}$ can effectively push the surface points towards their corresponding 2D coordinates by minimizing the overall distance. Since this EM-style optimization is prone to local-minima as we jointly optimize camera viewpoints, shapes and pose articulations, we find that a good initialization of 3D features is important. For this, we manually map each part index $i$ to one of the DINO clusters (4 in our experiments) and then initialize the 3D features $Q$ by assigning the average 2D cluster features to the corresponding 3D part vertices. This manual 3D part to 2D cluster assignment takes minimal effort since it is category-level (all instances share the same 3D parts). Alternatively, one could roughly initialize the camera viewpoints like in recent works such as NeRS [45] which allows the computation of initial $Q$ features from 2D features. Nonetheless, 2D-3D assignment is easier and faster compared to camera viewpoint initialization as 2D-3D assignment only needs to be done once per image ensemble whereas camera initialization is required for each instance in the ensemble. To make the optimization process fully automatic, we also propose to use simple heuristics for the 2D-3D part assignment using intuitions like animal head are usually above the torso and legs at the bottom. We refer to the LASSIE optimization with this heuristic-based 3D feature initialization as 'LASSIE-heuristic' and describe the details in the supplemental material.

**Comparison to related methods.** Our use of 2D-3D semantic consistency is closely related to prior works, UMR [24] and BANMo [42]. At a high-level, LASSIE and UMR both start from 2D part segmentations and then lift them onto 3D. A key difference is that LASSIE directly discovers parts in 3D using 2D feature consistency, whereas UMR uses 2D part consistency via canonical UV maps. More specifically, UMR uses 2D parts (not 2D features) and 2D UV maps as common canonical part representations. Instead, LASSIE directly optimizes 3D part features with feature based loss (not 2D parts) and LASSIE does not use UV maps like in UMR. We only use 2D parts for feature initialization and not during optimization. That is, we discover parts directly in 3D with skeleton constraints, without much reliance on intermediate 2D part discovery. Hence, LASSIE is less sensitive to the issues in 2D part segmentations compared to UMR. BANMo, on the other hand, learns a canonical feature embedding by enforcing the consistency between feature matching and geometric warping, which is made tractable by the dense temporal correspondence (optical flow) between video frames. The semantic loss in BanMo enforces the 2D-3D cycle consistency between 2D coordinates and canonical 3D points. Considering that our image ensemble is sparse and un-correlated, we coarsely initialize and regularize the 3D surface features at the part-level. The proposed semantic loss allows us to first localize the 3D parts then refine the detailed part shapes in our challenging setting.

**Regularization.** Our skeleton-based representation allows us to conveniently impose pose prior or regularization. Specifically, we calculate a pose prior loss $\mathcal{L}_{pose}$ to minimize the bone rotation

deviations from the resting pose as: $\mathcal{L}_{pose} = \sum_j \|R^j - \bar{R}\|^2$, where $R^j$ is the bone rotations of instance $j$ and $\bar{R}$ denotes the bone rotations of shared resting pose. We also impose a regularization $\mathcal{L}_{ang}$ on joint angles on the animal legs to limit their sideway rotations:

$$\mathcal{L}_{ang} = \sum_j \sum_{i \in \text{leg bones}} \|R_{i,y}{}^j\|^2 + \|R_{i,z}{}^j\|^2, \tag{5}$$

where $R_{i,y}$ and $R_{i,z}$ are the rotations of bone $i$ with respect to $y$ and $z$ axes respectively. Since the skeleton faces the $+z$ direction in the canonical space ($+x$ right, $+y$ up, $+z$ out), minimizing the $y$ and $z$-axis rotations essentially constrains the sideway movements of the bones. Note that both $\mathcal{L}_{pose}$ and $\mathcal{L}_{ang}$ are generic to all quadrupeds in our experiments, which we find crucial to avoid unrealistic poses. Finally, we apply common mesh regularization terms to encourage smooth surfaces, including the Laplacian loss $\mathcal{L}_{lap}$ and surface normal loss $\mathcal{L}_{norm}$. $\mathcal{L}_{lap}$ regularizes 3D surfaces by pushing each vertex towards the center of its neighbors, and $\mathcal{L}_{norm}$ encourages neighboring mesh faces to have similar normal vectors. Note that we apply regularization losses to each part surface individually since the part shapes should be primitive-like and usually do not have sharp or pointy surfaces.

**Optimization and textured outputs.** The overall optimization objective is:

$$\mathcal{L}_{mask} + \lambda_1 \mathcal{L}_{sem} + \lambda_2 \mathcal{L}_{pose} + \lambda_3 \mathcal{L}_{ang} + \lambda_4 \mathcal{L}_{lap} + \lambda_5 \mathcal{L}_{norm}, \tag{6}$$

where $\{\lambda_i\}$ are weighting hyper-parameters. We first pre-train the part prior MLP $\mathcal{F}^p$ with 3D geometric primitives and freeze it during the optimization on a given image ensemble. For each image ensemble of an animal species, we perform multi-stage optimization on the camera, pose, and shape parameters until convergence. That is, we update the camera viewpoints and fix the rest first, then optimize the bone transformations, and finally the latent part codes as well as part deformation MLPs. In each iteration, we first update the semantic features of 3D surfaces, then use the updated features to update 3D surfaces, forming an EM-style optimization. We implement the framework in PyTorch [33] and update all the learnable parameters using an Adam optimizer [16]. To obtain the final textured outputs, we densely sample vertices from the optimized neural surfaces and directly sample the colors of visible vertices from individual images via 2D projection, where the visibility information is obtained from naive rasterization. For the invisible (self-occluded) vertices, we assign the color by finding their left-right symmetries or nearest visible neighbors. More implementation details can be found in the supplemental material.

## 4   Experiments

**Datasets.** Considering the novel problem of sparse-image optimization, we select a subset of the Pascal-Part dataset [7] and collect web images of additional animal classes to evaluate LASSIE and related prior arts. The Pascal-Part dataset contains diverse images of several animal classes which are annotated with 2D part segmentation masks. We automatically select images of horse, cow, or sheep with only one object covering 1-50% area of the entire image, and find the keypoints by calculating the centers/corners of ground-truth part masks. In addition, for analysis on more diverse animal categories, we collect sparse image ensembles (with CC-licensed images) of some other animals from the internet and manually annotate the 2D keypoints for evaluation. The classes include quadrupeds like zebra, tiger, giraffe, elephant, as well as bipeds like kangaroo and penguin. We filter out the images where the animal body is heavily occluded or truncated, resulting in roughly 30 images per category. The LASSIE optimization and evaluations are performed on each image ensemble separately. The self-collected data does not contain personally identifiable information or offensive content, and will be released to public.

**Baselines.** Due to the lack of prior works on our problem setting (sparse image optimization for articulated animal shapes), we mainly compare LASSIE with learning-based mesh reconstruction methods. Among these methods, we find 3D Safari [47] and A-CSM [20] to be most comparable to LASSIE since they also model articulation for animal classes of our interest. Other recent mesh reconstruction methods, on the other hand, either cannot handle articulations [21, 24, 11, 36, 44] or assume different inputs [23, 40, 42]. Note that both 3D Safari and A-CSM are trained on large-scale image sets while LASSIE optimizes on a sparse image ensemble. To evaluate their methods on our dataset, we use the released models trained on the closest animal classes. For instance, the zebra model of 3D Safari and the horse model of A-CSM are used to evaluate on similar quadrupeds.

**Visual comparisons.** Fig. 3 shows the qualitative results of LASSIE against 3D Safari and A-CSM on the self-collected animal images. The 3D Safari [47] model is trained on zebra images and does

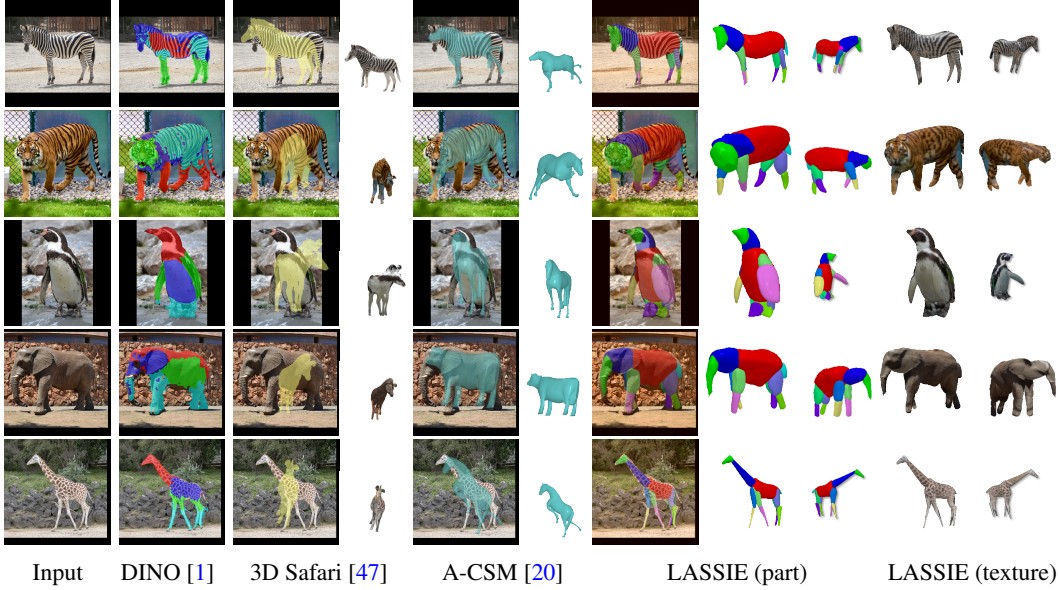

| Input | DINO [1] | 3D Safari [47] | A-CSM [20] | LASSIE (part) | LASSIE (texture) |

Figure 3: **Qualitative results on the in-the-wild image collections.** We show example results of the baselines as well as the part and texture reconstruction by LASSIE on the self-collected animal image ensembles. The results demonstrate the semantic consistency of discovered parts across diverse classes and the high-quality shape reconstruction that allows dense texture sampling.

Table 1: **Keypoint transfer evaluations.** We evaluate on all the source-target image pairs and report the percentage of correct keypoints under two different thresholds (PCK@0.1/ PCK@0.05).

| | Pascal-Part dataset | | | Our dataset | | | | | |
|---|---|---|---|---|---|---|---|---|---|
| Method | Horse | Cow | Sheep | Zebra | Tiger | Giraffe | Elephant | Kangaroo | Penguin |
| 3D Safari [47] | 71.8/ 57.1 | 63.4/ 50.3 | 62.6/ 50.5 | **80.8**/ 62.1 | 63.4/ 50.3 | 57.6/ 32.5 | 55.4/ 29.9 | 35.5/ 20.7 | 49.3/ 28.9 |
| A-CSM [20] | 69.3/ 55.3 | 68.8/ 60.5 | 67.4/ 54.7 | 78.5/ 60.3 | 69.1/ 55.7 | 71.2/ 52.2 | 67.3/ 39.5 | 42.1/ 26.9 | 53.7/ 33.0 |
| LASSIE w/o $\mathcal{L}_{sem}$ | 60.4/ 45.6 | 58.9/ 43.1 | 55.3/ 42.3 | 62.7/ 47.5 | 53.6/ 40.0 | 54.7/ 28.9 | 52.0/ 25.6 | 33.8/ 19.8 | 50.6/ 25.5 |
| LASSIE w/o $\mathcal{F}^p$ | 71.1/ 56.3 | 69.0/ 59.5 | 68.9/ 52.2 | 77.0/ 60.2 | 71.5/ 59.2 | 79.3/ 57.8 | 66.6/ 37.1 | 44.3/ 30.1 | 63.3/ 38.2 |
| LASSIE | **73.0/ 58.0** | **71.3/ 62.4** | **70.8/ 55.5** | 79.9/ **63.3** | **73.3/ 62.4** | **80.8/ 60.5** | **68.7/ 40.3** | **47.0/ 31.5** | **65.5/ 40.6** |
| LASSIE-heuristic | 72.1/ 57.0 | 69.5/ 61.1 | 69.7/ 55.3 | 78.9/ 61.9 | 71.9/ 60.0 | 80.4/ 60.1 | 66.5/ 38.7 | 45.6/ 30.9 | 60.8/ 37.6 |

not generalize well to other animal classes. A-CSM [20] assumes high-quality skinned model of an object category which enables detailed shapes of animal bodies. However, its outputs do not align well with the given 2D images. Our results demonstrate that LASSIE can effectively learn from sparse image ensemble to discover 3D articulated parts that are high-quality, faithful to input images, and semantically consistent across instances. Note that our 3D reconstructions also improve the 2D part segmentation from DINO-ViT feature clustering [1].

**Keypoint transfer metrics.** As there is no ground-truth 3D annotation in our datasets, we quantitatively evaluate using 2D keypoint transfer between each pair of images which is a standard practice [47, 20]. We map a given set of 2D keypoints on a source image onto the 3D part surfaces, and then project them to a target image using the optimized camera viewpoints, shapes and pose articulations. Since this keypoint transfer goes through the 2D-to-3D and 3D-to-2D mappings, a successful keypoint transfer requires accurate 3D reconstruction on both the source and target images. We calculate the percentage of correct keypoints (PCK) under two different thresholds: $0.1 \times \max(h, w)$ and $0.05 \times \max(h, w)$, where $h$ and $w$ are image height and width, respectively. Table 1 shows that LASSIE achieves higher PCK on the Pascal-part [7] images and most other classes in our in-the-wild image collections. We also show ablative results of LASSIE without using the semantic feature consistency loss $\mathcal{L}_{sem}$ or part prior MLP $\mathcal{F}^p$. Note that $\mathcal{L}_{sem}$ is essential to fit the skeleton and part surfaces faithfully to the input images, and $\mathcal{F}^p$ provides effective regularization on part surfaces to generate realistic shapes from various viewpoints. LASSIE-heuristic, by using

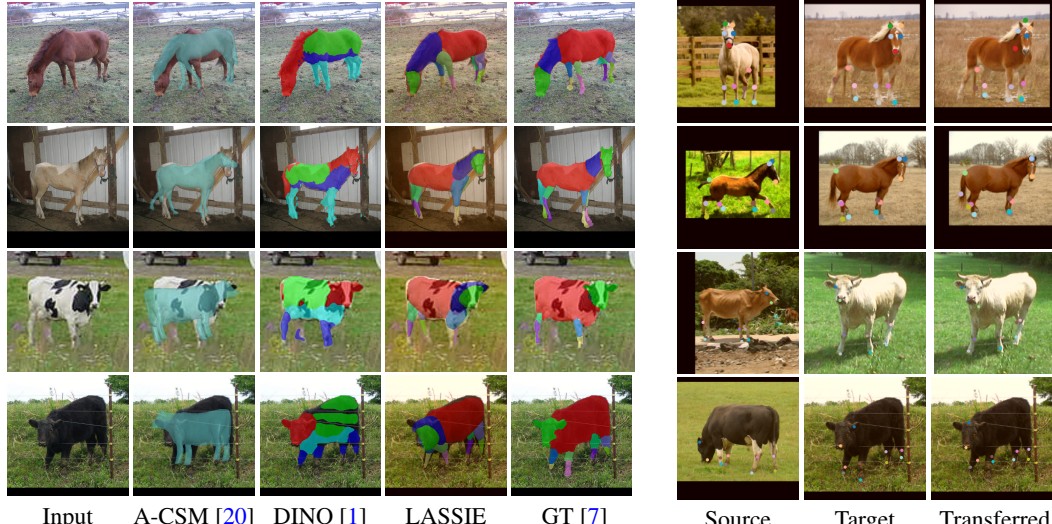

| Input | A-CSM [20] | DINO [1] | LASSIE | GT [7] |

| Source | Target | Transferred |

Figure 4: **2D segmentations.** We show the overall masks (A-CSM) and part masks (DINO clusters, LASSIE results, and Pascal-part GT) overlaid on the sample input images.

Figure 5: **Keypoint transfer results** using LASSIE from source to target images.

Table 2: **Quantitative evaluations on the Pascal-part images.** We report the overall foreground IOU, part mask IOU, and percentage of correct pixels (PCP) under dense part segmentation transfer between all pairs of source-target images.

| Method | Overall IOU | | | Part IOU | | | Part transfer (PCP) | | |
| --- | --- | --- | --- | --- | --- | --- | --- | --- | --- |
| | Horse | Cow | Sheep | Horse | Cow | Sheep | Horse | Cow | Sheep |
| SCOPS [13] | 62.9 | 67.7 | 63.2 | 23.0 | 19.1 | 26.8 | - | - | - |
| DINO clustering [1] | 81.3 | 85.1 | 83.9 | 26.3 | 21.8 | 30.8 | - | - | - |
| 3D Safari [47] | 72.2 | 71.3 | 70.8 | - | - | - | 71.7 | 69.0 | 69.3 |
| A-CSM [20] | 72.5 | 73.4 | 71.9 | - | - | - | 73.8 | 71.1 | 72.5 |
| LASSIE w/o $\mathcal{L}_{sem}$ | 65.3 | 67.5 | 60.0 | 29.7 | 23.2 | 31.5 | 62.0 | 60.3 | 59.7 |
| LASSIE w/o $\mathcal{F}^p$ | 81.0 | 86.5 | 85.0 | 37.1 | 33.7 | 41.9 | 76.1 | 75.6 | 72.4 |
| LASSIE | **81.9** | **87.1** | **85.5** | **38.2** | **35.1** | **43.7** | **78.5** | **77.0** | **74.3** |

automatic 3D feature initialization, achieves slightly lower PCK but still performs favorably against prior arts overall. Example visual results of keypoint transfer are shown in Fig. 5.

**2D overall/part IOU metrics.** Our part-based representation can generate realistic 3D shapes and can even improve 2D part discovery compared to prior arts. In Table 2 and Fig. 4, we show the qualitative and quantitative comparisons against prior 3D works as well as 2D co-part discovery methods of SCOPS [13] and DINO clustering [1]. For SCOPS [13] and DINO clustering [1], we set the number of parts $N_p = 4$ (which we find to be optimal for these techniques) and manually assign each discovered part to the best matched part in the Pascal-part annotations. Note that higher part IOU against Pascal-part annotations may not strictly indicate better part segmentation since the self-discovered parts need not correspond to human annotations. Compared to prior arts on articulated shape reconstruction, LASSIE results match the Pascal-part segmentation masks better and achieves higher overall IOU. On co-part segmentation, our 3D reconstruction captures the semantic parts in 2D while being more geometrically refined than DINO clustering. Moreover, our results can separate parts with similar semantic features, *e.g.*, 4 animal legs, and thus resulting in much higher part IOU.

**Part transfer.** While keypoint transfer and part IOU are commonly used in the literature, we observe that they are either sparse, biased towards certain body parts (*e.g.*, animal faces), or not directly comparable due to part mismatch. To address this issue, we propose a part transfer metric for better evaluation of 3D reconstruction consistency across image ensemble. This part transfer metric is similar to keypoint transfer but uses part segmentations. That is, we densely transfer the part

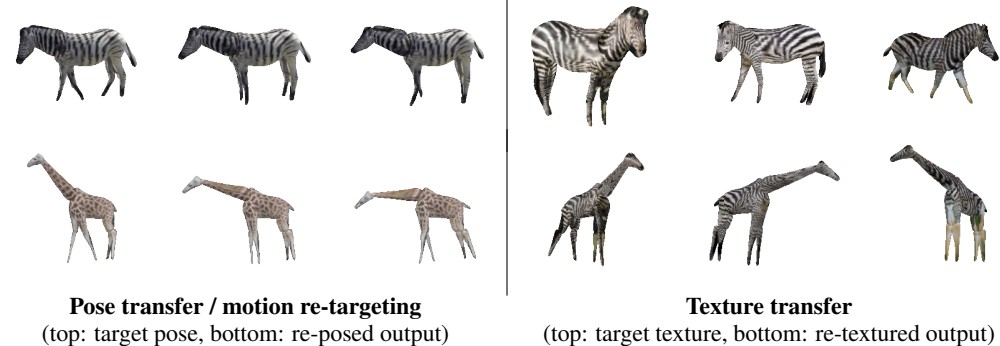

| **Pose transfer / motion re-targeting** | **Texture transfer** |
|:---:|:---:|
| (top: target pose, bottom: re-posed output) | (top: target texture, bottom: re-textured output) |

Figure 6: **Applications of neural part surfaces.** We use the LASSIE results of zebra images (top) as target and apply pose transfer (bottom left) and texture transfer (bottom right) to the giraffe shapes. These applications demonstrate the realistic part surfaces discovered by LASSIE.

segmentation in a source image to a target image through forward and backward mapping between the 2D pixels and the canonical 3D surfaces. A pixel is considered transferred correctly if and only if it is mapped to the same 2D part in the target image as where it belongs in the source image. Similar to PCK in keypoint transfer evaluation, we calculate the percentage of correct pixels (PCP) for each source-target pair. Since the part segmentation only covers the visible surfaces and not the occluded regions, we ignore the pixels mapped to a self-occluded surface when calculating the PCP metric. The quantitative results are shown in Table 2, which demonstrate the favorable performance of LASSIE against prior arts.

**Applications.** Our 3D neural part representation not only produces more faithful and realistic shapes but also enables various applications like part manipulation, texture transfer, motion re-targeting, etc. Specifically, we can generate a new shape by swapping or interpolating certain parts, transfer the sampled surface texture from one object class to another using the same base skeleton, or specify the bone rotations to produce desirable re-posed shape or animation. We show some example results of pose transfer and texture transfer between different animal classes in Fig. 6.

**Limitations.** LASSIE relies heavily on the 3D skeleton and strong pose and shape regularization. As the first attempt to address this novel and highly ill-posed problem, we find the proposed pose and shape constraints essential to produce good quality results and avoid unrealistic outputs. Moreover, LASSIE can not handle image collections with heavy truncation, occlusions, or extreme pose variations since we leverage self-supervisory DINO-ViT features on sparse image ensemble as our main supervision. For instance, partial-body or noisy DINO features can cause ambiguities in camera pose and part localization. Finally, we observe that LASSIE currently struggles with a) highly articulated parts like elephant trucks and b) fluffy animals that appear with more instance variations and ambiguous articulations.

## 5  Conclusion

In this paper, we study a novel and practical problem of articulated shape optimization of animals from sparse image collections in-the-wild. Instead of relying on pre-defined shape models or human annotations like keypoints or segmentation masks, we propose a neural part representation based on a generic 3D skeleton, which is robust to appearance variations and generalizes well across different animal classes. In LASSIE, our key insight is to reason about animal parts instead of whole shape as parts allow imposing several constraints. We leverage self-supervisory dense ViT features to provide both silhouette and semantic part consistency for supervision. Both quantitative and qualitative results on Pascal-Part and our own in-the-wild image ensembles show that LASSIE can effectively learn from few in-the-wild images and produce high-quality results. Our part-based representation also enables various applications such as pose interpolation, texture transfer, and animation. We hope that this work will facilitate future research efforts on learning, inferring, and manipulating articulated shapes from image ensembles in the wild.

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
