# *Supplementary Material for*
# LASSIE: Learning Articulated Shapes from Sparse Image Ensemble via 3D Part Discovery

Chun-Han Yao[1]    Wei-Chih Hung[2]    Yuanzhen Li[3]    Michael Rubinstein[3]
Ming-Hsuan Yang[134]    Varun Jampani[3]

[1]UC Merced    [2]Waymo    [3]Google Research    [4]Yonsei University

In this supplementary document, we present the implementation details, model analyses, and additional results of our method. We also provide a short video to explain our framework with illustrations and visual results. Given the 3D nature of our results, we recommend readers to see the supplementary video for better visualization of our 3D reconstructions.

## 1 Implementation Details

We implement the LASSIE framework using PyTorch [8], and optimize all parameters using an Adam optimizer [4]. For an image ensemble containing 30 images, the overall optimization takes roughly 10 minutes on a single GTX 1080 GPU.

### 1.1 Notation Table

Table 1: **Notations.** For key variables in the paper, we list the symbol, variable name, state space, and whether the variable is instance-specific or shared within an animal class.

| Symbol | Variable name | State space | Instance-specific |
|---|---|---|---|
| $\pi = (R_0, t_0)$ | Camera viewpoint | $(\mathbb{R}^{3\times3}, \mathbb{R}^3)$ | ✓ |
| $R_i$ | Global rotation of part $i$ | $\mathbb{R}^{3\times3}$ | ✓ |
| $\bar{R}_i$ | Resting rotation of part $i$ | $\mathbb{R}^{3\times3}$ | |
| $s_i$ | Part scaling of part $i$ | $\mathbb{R}$ | |
| $\mathbf{e}_i$ | Primitive shape code of part $i$ | $\mathbb{R}^d$ | |
| $\mathcal{F}^p$ | Primitive MLP | MLP | |
| $\mathcal{F}_i^\Delta$ | Part deformation MLP of part $i$ | MLP | |

### 1.2 ViT Feature Extraction

To extract the semantic features from images, we use a self-supervised ViT (DINO-ViT) trained via a self-distillation approach [2]. Similar to [1, 10], we observe that the *key* features is a robust visual descriptor and can be use to find structural or semantic correspondence between images. Specifically, we extract the keys from the last layer of DINO given an input image of size $512 \times 512$, resulting in a feature map of size $64 \times 64$. Likewise, we extract the *class* tokens and use their average attention map as a saliency estimation. We then collect and cluster the features of salient image patches by thresholding the saliency scores. The feature clustering is done by an off-the-shelf K-means algorithm, using number of clusters $c = 4$. Finally, we obtain a pseudo ground-truth object silhouette $\hat{M}$ by thresholding the minimum feature distance to cluster centroids. We apply PCA on the extracted keys to reduce from 128-dimensional to $f = 64$ and use them to more efficiently calculate semantic consistency loss. Figure 1 visualizes the DINO feature extraction steps.

36th Conference on Neural Information Processing Systems (NeurIPS 2022).

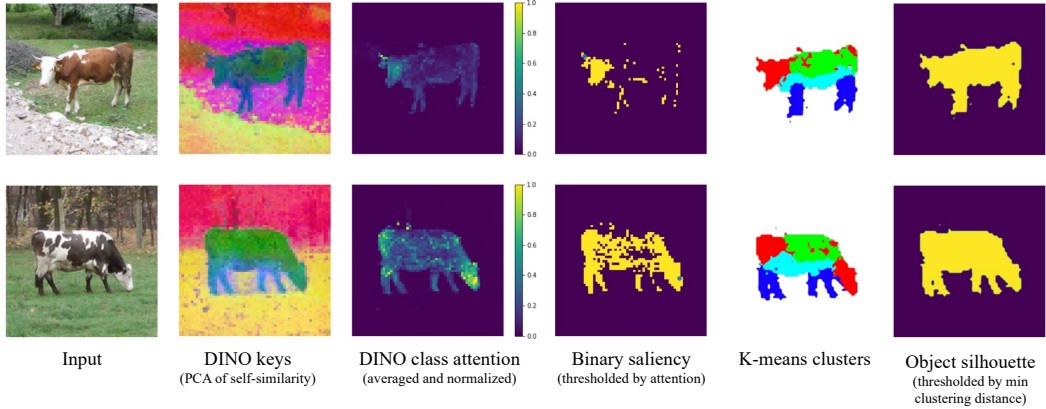

| Input | DINO keys
(PCA of self-similarity) | DINO class attention
(averaged and normalized) | Binary saliency
(thresholded by attention) | K-means clusters | Object silhouette
(thresholded by min
clustering distance) |

Figure 1: **Exploiting self-supervised DINO-ViT features.** We visualize the keys, saliency estimation, K-means clustering results, and pseudo ground-truth silhouette.

## 1.3 Generic 3D Skeletons

We use two generic 3D skeletons in our experiments, one for quadrupeds (four-legged) and another for bipedal (two-legged) animals. Both skeleton has 16 joints and 15 bones, from which we build 15 body parts including head, neck, torso, tail, 4 upper legs, 4 middle legs, and 4 lower legs. Figure 2 illustrates the skeleton for quadrupeds. The bipedal skeleton has a similar structure but different initial joint coordinates since they stand on two back legs. The initial joint coordinates are defined in the canonical space $(0, 1)^3$ relative to the root joint at $(0, 0, 0)$. The category-specific skeleton is optimized by updating the bone length scaling $s \in \mathbb{R}^b$ and bone rotations at resting pose $\bar{R} \in \mathbb{R}^{b \times 3}$.

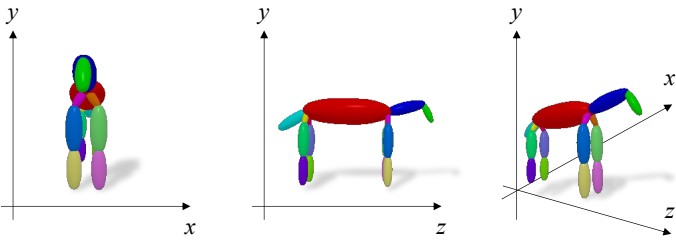

Figure 2: **Generic 3D skeleton for quadrupedal animals.** We define a skeleton by specifying the initial joint coordinates in the canonical space and the bone transformations at resting pose.

## 1.4 Neural Part Surfaces

We represent our neural surfaces in a similar fashion as NeRS [13] except that we decompose a complex shape into multiple parts with simple geometry. For each part, we uniformly sample a set of surface coordinates $X = \{(x, y, z) | x^2 + y^2 + z^2 = 1\}$ from a unit sphere and decode their coordinate/deformation in the canonical space via an MLP. Following [7, 9, 13], we apply positional encoding on the surface coordinates to facilitate learning high frequency functions. As shown in Figure 3, the primitive MLP and part MLPs adopt a similar architecture as NeRS, *i.e.*, three fully-connected layers with instance normalization and Leaky ReLU activation for middle layers. The surface coordinates $X$ are randomly jittered during MLP optimization to ensure a smooth output surface. We symmetrize the canonical-space outputs with respect to the $x = 0$ plane since animal bodies are left-right symmetric. Then, we scale, translate, and rotate the parts by their corresponding bone transformations to obtain the instance-specific articulation. For rendering, we combine all parts by concatenating the vertices and faces defined in the unit sphere. Given the optimization results, the surface texture can be directly sampled from each input image and symmetrized.

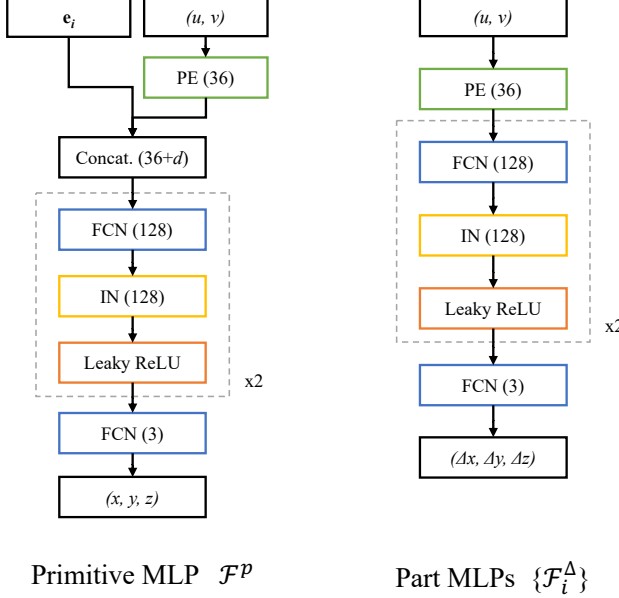

Primitive MLP $\mathcal{F}^p$       Part MLPs $\{\mathcal{F}_i^\Delta\}$

Figure 3: **MLP architecture.** We show the architecture diagrams for primitive MLP and part MLPs. The $(u, v)$ coordinates are parametrized as 3D coordinates sampled from a unit sphere. We use positional encoding (PE) with 6 sine and cosine bases, thus mapping 3D to 36D. Each middle block is composed of a fully-connect layer (FCN), instance normalization [11] (IN), and LeakyReLU.

## 1.5 Part Prior Learning

To learn a latent shape space for simple parts, we train a Variational Auto-Encoder (VAE), called Part VAE. Since each part should cover the corresponding skeleton bone, we train Part VAE with linear combinations of 3D geometric primitives like ellipsoids, cylinders, and cones. We illustrate part prior learning in Figure 4. After pre-train Part VAE, we discard the primitive encoder and fix the primitive decoder MLP during LASSIE optimization. As such, since the primitive MLP is conditioned by the latent part codes, we can produce different part shapes by simply optimizing the part codes. The part prior learning is similar to [12] for rigid objects, but our primitive decoder is a conditional neural surface built for skeleton-based articulated parts.

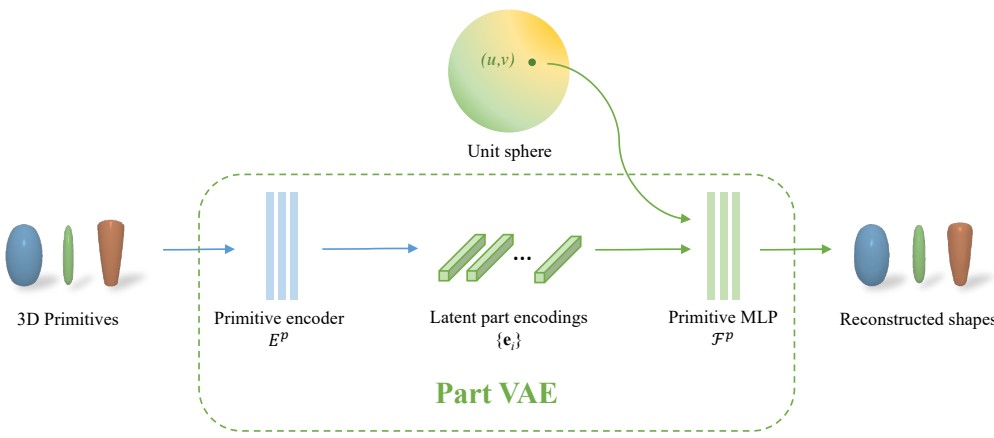

Figure 4: **Learning part prior with Part VAE.** We train the Part VAE with 3D primitive shapes. The reconstruction is supervised by an L2 loss and mesh regularization. The latent encodings are regularized by a KL-divergence loss.

## 1.6 Optimization Pseudo-code

We provide the pseudo-code below to clarify the optimization process. The overall optimization include 4 main stages: 1) camera, 2) camera and pose, 3) shape, and 4) all parameters. The parameters are optimized using the corresponding losses in each stage. In each iteration, we first update the semantic features of 3D surfaces, then use the updated features to update 3D surfaces, forming an EM-style optimization. This EM-style semantic and geometry optimization is crucial since it can progressively refine the 3D surface features given better pose and shape fitting, then in return provide more accurate surface-pixel correspondence.

---

**Algorithm 1  LASSIE multi-stage optimization**

---

**Parameters:** resting part rotations $\bar{R}$ , bone length scaling $\{s_i\}$, part rotation $\{R_j\}$, camera viewpoints $\{\pi_j\}$, latent part codes $\{e_i\}$, part deformation MLPs $\{\mathcal{F}_i^\Delta\}$ ($i$: part index, $j$: instance index).
**Losses:** mask IOU loss $\mathcal{L}_{mask}$, semantic consistency loss $\mathcal{L}_{sem}$, pose deviation loss $\mathcal{L}_{pose}$, part angle prior $\mathcal{L}_{ang}$, Laplacian regularization $\mathcal{L}_{lap}$, surface normal loss $\mathcal{L}_{norm}$.

 1: Stage 1: Optimize $\{\pi_j\}$ using $\mathcal{L}_{sem}$ until convergence
 2: Stage 2: Optimize $\{\pi_j\}$, $\{s_i\}$, $\bar{R}$, and $\{R_j\}$ using $\mathcal{L}_{sem}, \mathcal{L}_{pose}, \mathcal{L}_{ang}$ until convergence
 3: Stage 3: Optimize $\{e_i\}$ and $\{\mathcal{F}_i^\Delta\}$ using $\mathcal{L}_{mask}, \mathcal{L}_{sem}, \mathcal{L}_{lap}, \mathcal{L}_{norm}$ until convergence
 4: Stage 4: Optimize all parameters using all losses until convergence

---

**Algorithm 2  EM-style semantic and geometric optimization**

---

**Parameters:** 3D surface features $Q$.
**Losses:** semantic consistency loss $\mathcal{L}_{sem}$.

 1: **repeat**
 2:     **E-step**: Update 3D surface features $Q$ by rendering the neural surfaces on each image, finding the nearest pixel for each 3D point, and averaging the corresponding image features.
 3:     **M-step**: Optimize neural surfaces using the updated $Q$ in $\mathcal{L}_{sem}$ (Eq. 3 in manuscript). Note that the M-step also involves updating other parameters with different losses depending on the optimization stage.
 4: **until** end of optimization stage

---

## 2 Ablative Study

### 2.1 Heuristic Part Mapping for 3D Feature Initialization

To initialize the 3D vertex features $Q$ for semantic consistency loss, we manually assign each skeleton part to a DINO-ViT feature cluster. Although the manual assignment takes minimal effort, we can alternatively design simple heuristics for part mapping and make the optimization framework fully automatic. For instance, we assign the leg parts to a cluster whose average vertical (Y-axis) coordinate is the lowest. Likewise, we map head and neck to the cluster with highest Y-coordinate. Then, we assign the rest (torso and upper legs) to the remaining clusters. We observe that this heuristic mapping is effective and generic to all animal classes in our experiments. We show the quantitative results of heuristic part mapping, denoted as LASSIE-heuristic, in Table 2.

Table 2: **Keypoint transfer evaluations (PCK@0.1).** LASSIE-heuristic leads to lower PCK than LASSIE, but still performs favorably against prior arts.

| Method | Zebra | Tiger | Giraffe | Elephant | Kangaroo | Penguin | Mean |
|---|---|---|---|---|---|---|---|
| 3D Safari [14] | **80.8** | 63.4 | 57.6 | 55.4 | 35.5 | 49.3 | 57.0 |
| A-CSM [6] | 78.5 | 69.1 | 71.2 | 67.3 | 42.1 | 53.7 | 63.7 |
| LASSIE | 79.9 | **73.3** | **80.8** | **68.7** | **47.0** | **65.5** | **69.2** |
| LASSIE-heuristic | 78.9 | 71.9 | 80.4 | 66.5 | 45.6 | 60.8 | 67.4 |

### 2.2 ViT Feature Clustering

In Figure 5, we compare the clustering results with different number of clusters $c$ and with/without applying a multi-label conditional random field (CRF) [5] for post-processing. We observe that using more number of clusters ($c = 16$ is the same number of parts in our skeleton definition) does not lead to better semantic part clusters and tends to over segment the animal bodies. For instance, the torso and head regions are clustered into many small pieces while the 4 legs are still in the same semantic cluster. Moreover, although applying CRF can produce smoother part segmentation and visually pleasing results, it is not necessary in the LASSIE optimization framework and sometime produce worse results on animal legs. In our experiments, we find that $c = 4$ without CRF leads to reasonable results on all animal classes while having little computational overhead.

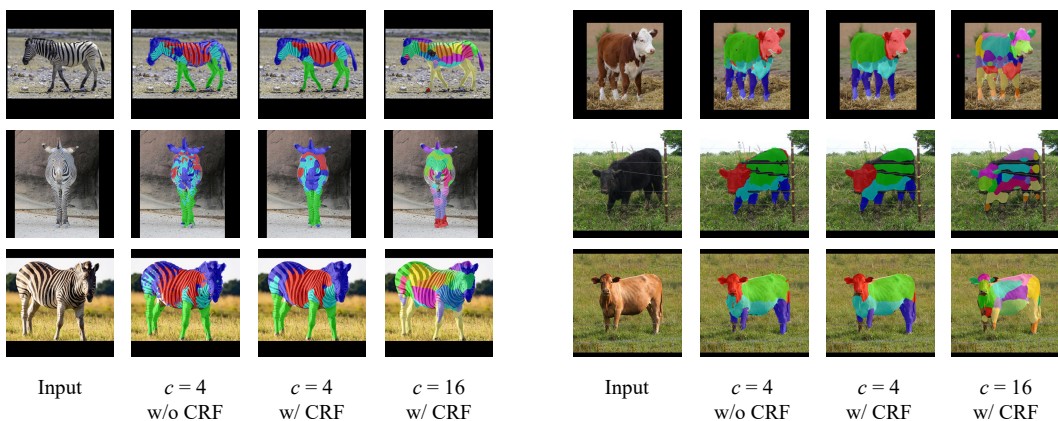

| Input | $c = 4$
w/o CRF | $c = 4$
w/ CRF | $c = 16$
w/ CRF | Input | $c = 4$
w/o CRF | $c = 4$
w/ CRF | $c = 16$
w/ CRF |

Figure 5: **Ablative comparisons on DINO feature clustering.** Using $c = 16$ tends to over-segment the animal bodies, and CRF sometimes produces less accurate silhouettes in order to smooth the segmentation masks. Therefore, we use $c = 4$ without CRF for LASSIE optimization.

## 2.3 Number of Images

In Figure 6, we show ablative comparisons of using different number of images for optimization.

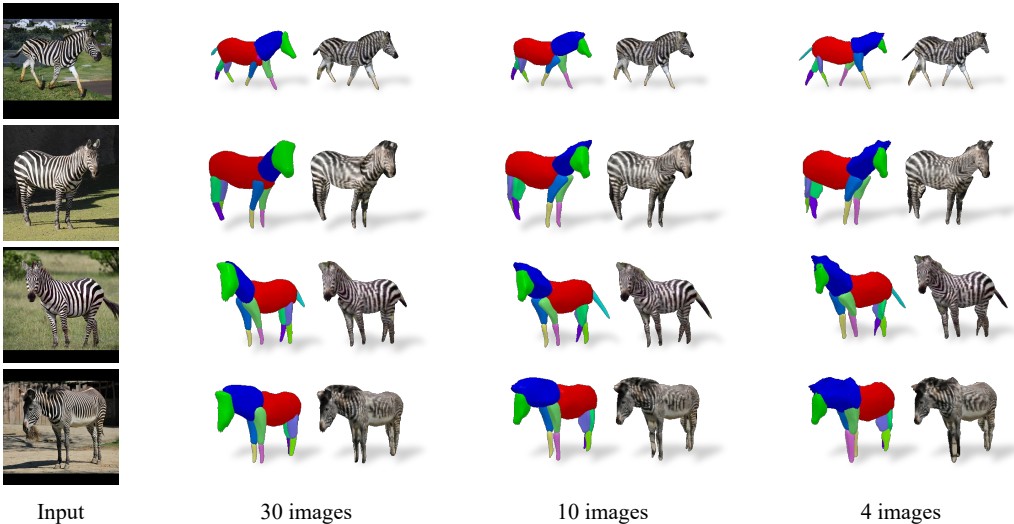

Figure 6: **Various number of images for optimization.** We show that LASSIE can still discover meaningful parts and produce good 3D reconstruction using a smaller-scale image ensemble ($n = 4$ or $n = 10$) if it contains diverse viewpoints and poses. However, more images ($n = 30$) result in more accurate part shapes.

## 2.4 Neural Part Surface v.s. Fixed-resolution Mesh

We compare the qualitative results with and without using the neural part representation in Figure 7. As a baseline, we use a pre-defined deformable mesh with 642 vertices and 1280 faces to represent each part, similar to [12] for rigid objects.

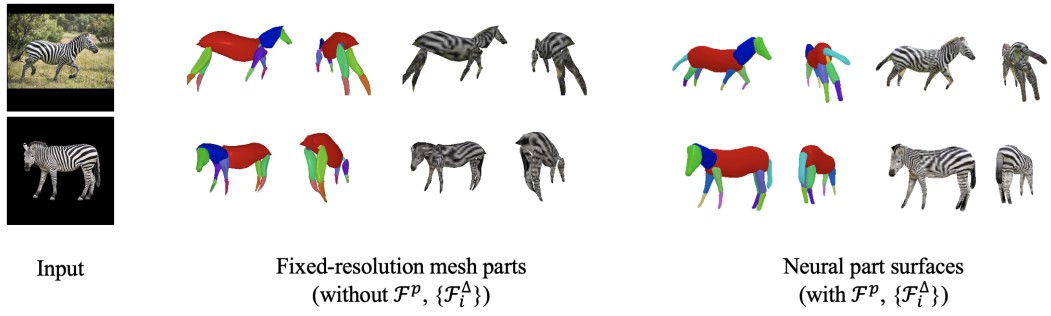

Figure 7: **Ablative comparisons of different 3D part representations.** We observe that using a pre-defined mesh for each part produces irregular shapes and less detailed texture since it is limited by the resolution (number of vertices and faces).

## 2.5 With/without Primitive Part Prior

To demonstrate the effectiveness of our primitive part prior learned from Part VAE, we show qualitative results without using $\mathcal{F}^p$ in Figure 8. Without $\mathcal{F}^p$, each part is directly represented by its individual part MLP $\mathcal{F}_i^\triangle$, and thus not constrained by the latent shape space.

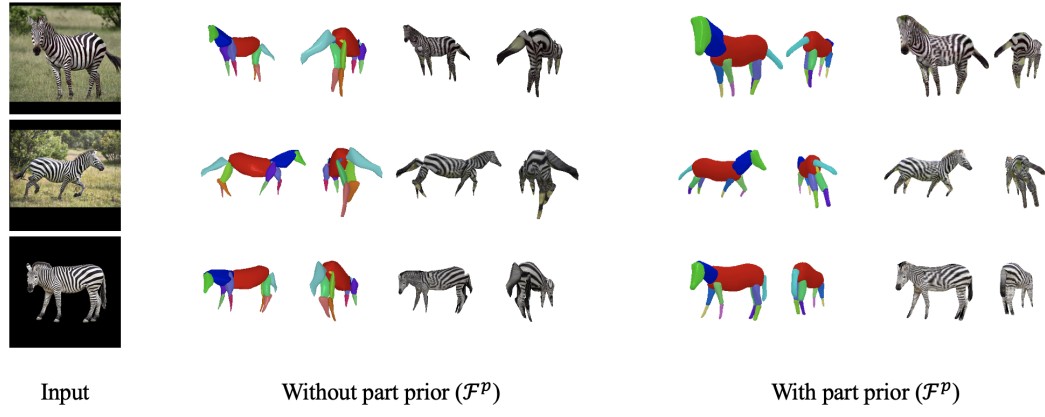

Input        Without part prior ($\mathcal{F}^p$)        With part prior ($\mathcal{F}^p$)

Figure 8: **Ablative comparisons of primitive part prior.** Without $\mathcal{F}^p$, the part shapes tend to have undesired sharp or thin structure, which are more visible from a novel view.

# 3 Datasets and Code

## 3.1 Dataset and Code Licenses

In our experiments, we make use of the publically available Pascal-part dataset [3] (http://roozbehm.info/pascal-parts/pascal-parts.html) as well as the released source code or models of the following methods:

- A-CSM [6]: https://github.com/nileshkulkarni/acsm/blob/master/LICENSE (Apache License 2.0)
- 3D Safari [14]: https://github.com/silviazuffi/smalst/blob/master/LICENSE.txt (MIT-License)
- NeRS [13]: https://github.com/jasonyzhang/ners/blob/main/LICENSE (BSD 3-Clause License)
- DINO-ViT [2]: https://github.com/facebookresearch/dino/blob/main/LICENSE (Apache License 2.0)
- DINO clustering [1]: https://github.com/ShirAmir/dino-vit-features/blob/main/LICENSE (MIT-License)

## 3.2 Dataset Statistics

We show the dataset statistics in Table 3, including the number of images $n$ per class for optimization and the number of source-target pairs for keypoint/part transfer evaluation. Note that the number of source-target pairs is $n \times (n-1)$ since we exhaustively use every image as source or target.

Table 3: **Dataset statistics.** We use all images in each class for optimization and evaluation on all source-target pairs of images.

| Class | Pascal-Part dataset [3] | | | Our image ensemble | | | | | |
|---|---|---|---|---|---|---|---|---|---|
| | Horse | Cow | Sheep | Zebra | Tiger | Giraffe | Elephant | Kangaroo | Penguin |
| # images | 30 | 30 | 16 | 30 | 28 | 30 | 30 | 30 | 32 |
| # source-target pairs | 870 | 870 | 240 | 870 | 756 | 870 | 870 | 870 | 992 |

## 3.3 Example Images

In Figure 9, we show several images sampled from each animal class to demonstrate that LASSIE can effectively learn from such sparse but diverse image ensemble.

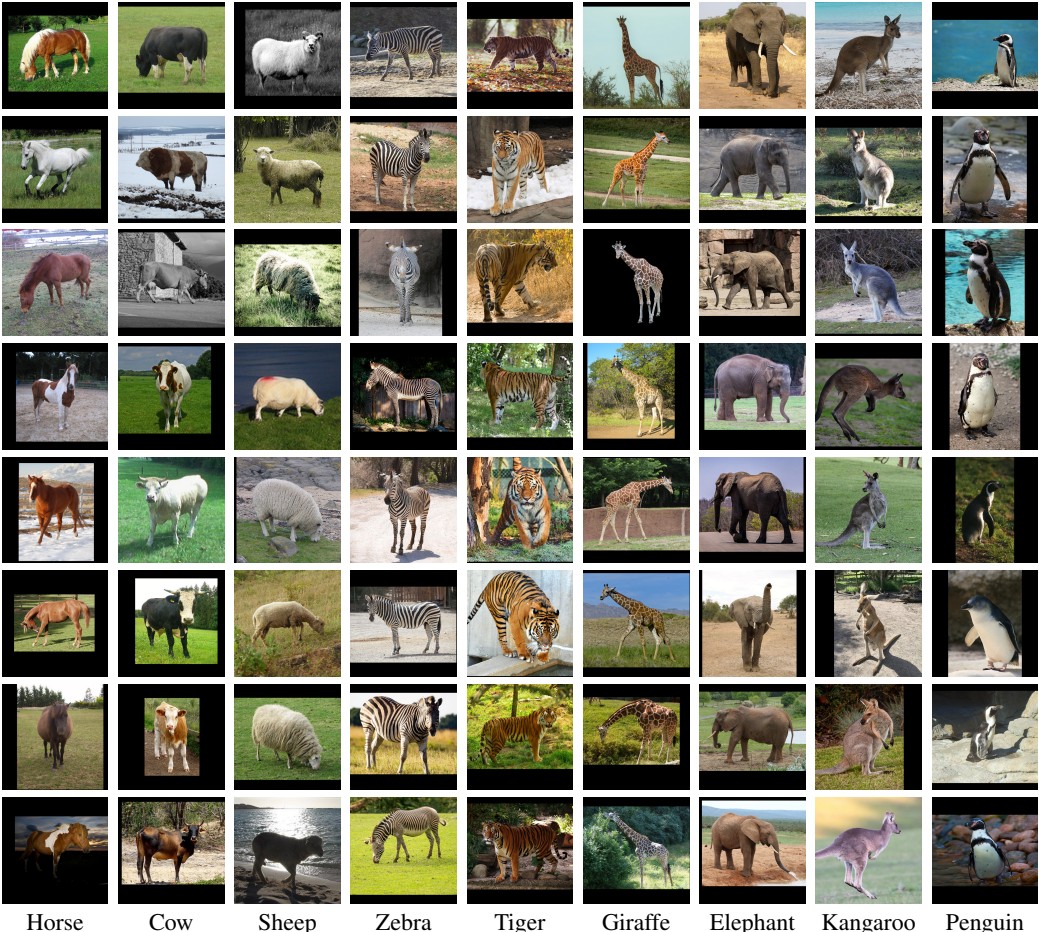

Horse     Cow     Sheep     Zebra     Tiger     Giraffe     Elephant     Kangaroo     Penguin

Figure 9: **Example images.** We randomly sample 8 image per animal class to show the diversity of image ensemble in our experiments.

# 4 Additional Results

## 4.1 All Animal Classes

We show more qualitative results on all animal classes in Figures 10 and 11 to demonstrate that LASSIE can be generalized to diverse articulated shapes.

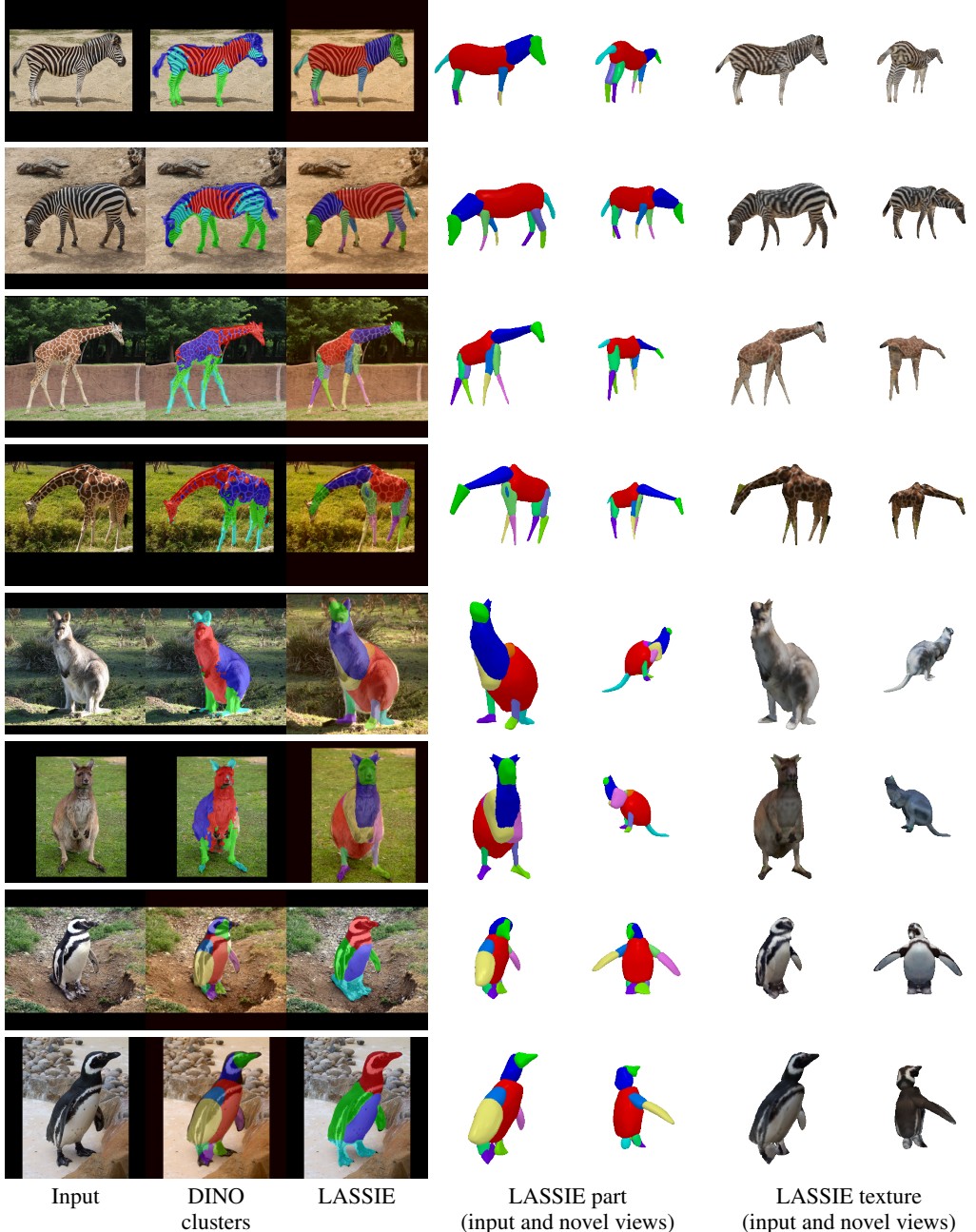

| Input | DINO clusters | LASSIE | LASSIE part (input and novel views) | LASSIE texture (input and novel views) |

Figure 10: **Additional qualitative results.** We show that LASSIE is able to model a wide range of animals, such as zebra, giraffe, kangaroo, and penguin (from top to bottom).

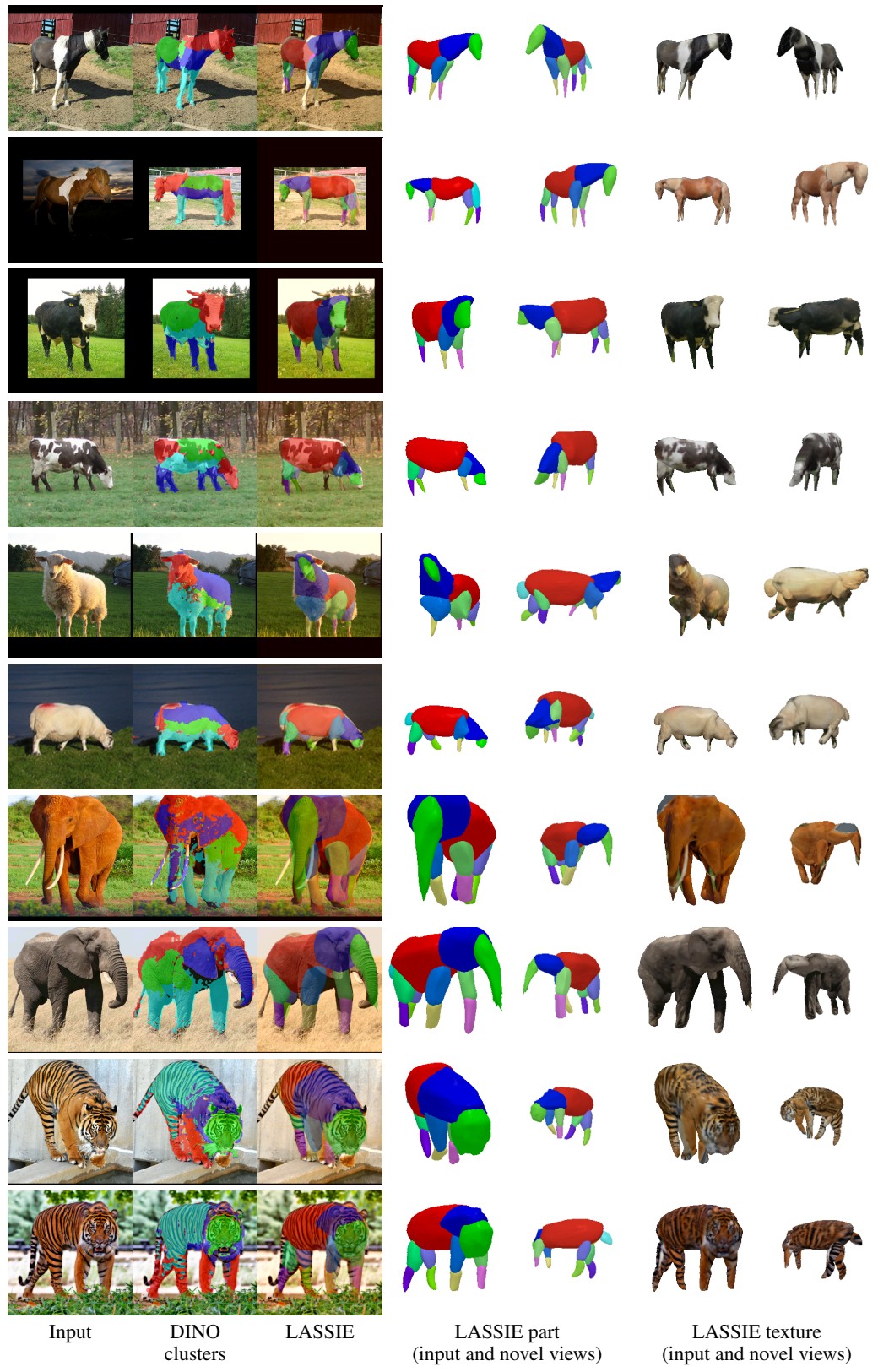

Input    DINO    LASSIE      LASSIE part      LASSIE texture
       clusters       (input and novel views)    (input and novel views)

Figure 11: **Additional qualitative results.** We show that LASSIE is able to model a wide range of animals, such as horse, cow, sheep, elephant, and tiger (from top to bottom).

## 4.2 Failure Cases

Finally, we show some failure cases in Figure 12. Since LASSIE does not use any image-level annotations for optimization, its performance is dependent on the self-supervised DINO features. Concretely, if the DINO clustering results are inaccurate, noisy, or ambiguous, then LASSIE might perform worse on those images. Common causes of such situation include ambiguous global pose, extreme articulations, and partial observations (occlusion or truncation).

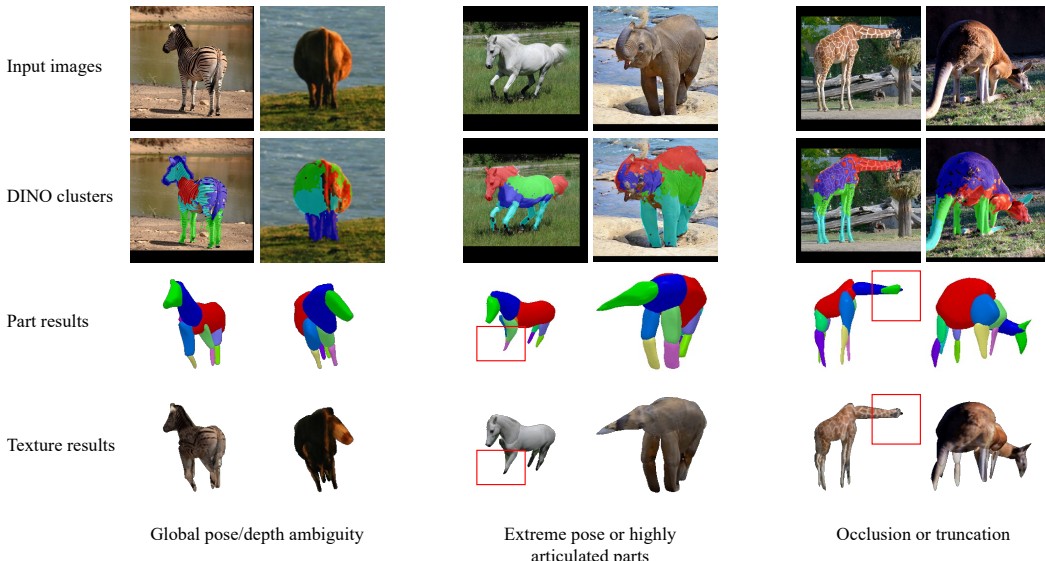

Figure 12: **Example failure cases.** LASSIE sometimes fail on images with ambiguous global pose/depth (columns 1-2), extreme articulations (columns 3-4), and partial observations (columns 5-6) caused by occlusion or truncation. Such cases can be improved by using rough camera/2D pose estimation and sparse keypoint annotations, which we do not assume for practicality.