# OpenReview forum: "LASSIE: Learning Articulated Shapes from Sparse Image Ensemble via 3D Part Discovery"
_NeurIPS.cc/2022/Conference — NeurIPS 2022 Accept_

### Official Review · Reviewer_mH51 · 2022-06-15

**Rating:** 7
**Confidence:** 4
**Soundness:** 4 excellent
**Presentation:** 3 good
**Contribution:** 3 good

**Summary:**

The submission propose Lassie to obtain articulated 3D reconstructions of a skeleton-based animal class (shown are quadrupeds and bipeds) from an image collection of 20-30 images of different individual animals. Given a fairly generic animal skeleton (16 vertices and 15 edges), it learns a 3D part for each edge/bone. The 3D part is represented by regressing 3D offsets for a 3D unit sphere via a coordinate-based MLP. The shapes of the part are shared across instances, only a similarity transform is applied based on the skeleton of each instance. The method requires only little outside knowledge: It clusters DINO-ViT [5] image features to obtain a very rough 2D part annotation of the images, and it requires manual matching between these part annotations and the skeleton parts. The optimization determines instance-specific skeleton parameters and class-specific part parameters. The main loss is a novel semantic matching loss that has some similarity to the one used in BanMo since it transfers image features to the 3D reconstruction and then tries to match the reconstruction to the images using those features.

There is no prior work that tackles this problem setting. Comparisons to A-CSM and 3D Safari show better results for Lassie, in several distinct settings.

**Questions:**

Questions

* The semantic loss is interesting. It would be nice to see a discussion on how it differs from BanMo's loss (see summary above).
* I would like the authors to comment on why comparisons against CMR, CSM, U-CMR, VMR, etc. (see Weaknesses) is not necessary, e.g. because there is simply no code.
* (Almost?) all methods that follow CMR (reconstructing animals/dynamic objects from image ensembles) evaluate on the CUB-200-2011 birds dataset. Why is there no evaluation on birds? Is there too much variablility? A-CSM [17], against which the submission compares, also evaluates on this dataset.
* The checklist states that error bars are provided. But I do not see any numbers or any other indication that multiple runs/seeds were used anywhere. Am I overlooking something or is this answer in the checklist wrong?

* While not crucial, the ablation on the number of images per animal in supplemental Sec. 2.2 uses the zebra class, which seems like a quiet easy class (see e.g. Fig. 9 in the supplement; all zebras have a pretty similar pose). I am curious how well a reduced number of images would work for giraffes or kangaroos or maybe horses, especially when keeping diverse poses and appearances in that subset of images.

Suggestions (do not need to be addressed in the rebuttal but should be considered for a revised version of the paper):

* I am not really satisfied with saying that no annotations are used (e.g. L.166). The DINO features give rise to 2D part segmentation masks. DINO is self-supervised, but still.
* While I find the evaluation sufficient already, results on humans would be interesting.
* The conclusion claims that no human annotations (L.342f) is used, but that's not correct (L.231ff). Please make sure that claims about annotations are precise everywhere.
* Please provide the mathematical formulation of L_lap and L_norm (and all other terms that might not be defined) in the supplementary material.
* In Fig. 10 and 11 in the supplemental document, please make the novel views of the same size as the input views.
* The current uniformly colored visualization of the parts shows coarse-level correspondences across instances. It could be interesting to color each part with some kind of pattern (e.g. a rainbow or color gradient) to visualize fine-level correspondences.
* L.167f says that the instances can have "varying texture (appearance) due to pose, camera angle, and lighting". What about the individual identity, i.e. different individuals have different fur? The method seems to not have an issue with it, I think it should be added to that list.
* L.256ff state that alternative optimization of camera, pose, and shape is used. Does that mean that the camera is optimized at the beginning of training and then never again? Or is it camera, then pose, then shape, then camera, then pose, then shape, etc.? If it's the former, maybe calling it training stages instead of alternative optimization would be more descriptive.
* What's the value of m in L.215?
* Please render the images in a higher resolution. They are quite pixelated already.

**Limitations:**

The checklist states that potential negative societal impact is discussed. While I don't see any such potential negative impact, I don't believe that that is discussed anywhere in the paper, despite what the checklist says.

Limitations are discussed, including failure cases in the supplement.

**Strengths And Weaknesses:**

=====STRENGTHS=====

The problem setting is interesting and it is remarkable that the proposed method takes only 10 minutes on a consumer GPU to obtain quite decent articulated reconstructions from only 20-30 images of an animal class. This opens up the door for a lot of follow-up work and raises a bunch of interesting questions, although unfortunately code will not be released.

While there are an MLP for the part shape and DINO image features, the majority of the method is not much concerned with neural networks, which is a nice change of pace.

The method is not exactly simple but also far from being convoluted. Its complexity seems appropriate to me and the design choices are argued for in the paper.

The evaluation is done well, including additional experiments in the supplement.

The paper is well written. The presentation is nice, including clear tables and figures.

I appreciate the name of the method.

=====WEAKNESSES=====

There are no significant weaknesses I can see. I do have a number of questions and suggestions though, see below.

Minor writing issues:
* L.12: "We" should be lower case
* L.30: "a" -> "the"
* L.146: "k" -> "b"
* L.286: sentence ends in "which"

Related Work:

* Missing related works that are from the same line of work as A-CSM [17]: Kanazawa et al. Learning category-specific mesh reconstruction from image collections (CMR; ECCV 2018) and its follow-up works, see e.g. Table 1 in Wu et al. DOVE: Learning Deformable 3D Objects by Watching Videos (arxiv preprint 2021):

- Kulkarni et al. Canonical surface mapping via geometric cycle consistency (CSM; ICCV 2019),
- Goel et al. Shape and viewpoints without keypoints (U-CMR; ECCV 2020),
- Li et al. Self-supervised single-view 3D reconstruction via semantic consistency (UMR; ECCV 2020),
- Li et al. Online adaptation for consistent mesh reconstruction in the wild (VMR; NeurIPS 2020),
- Kokkinos et al. Learning monocular 3D reconstruction of articulated categories from motion (CVPR 2021),
- Kokkinos et al. To The Point: Correspondence-driven monocular 3D category reconstruction (NeurIPS 2021)

* Less crucial, several animal shape estimation methods that use SMAL or similar statistical models are not cited:

- Biggs et al. Creatures great and SMAL: Recovering the shape and motion of animals from video (ACCV 2018)
- Biggs et al. Who left the dogs out: 3D animal reconstruction with expectation maximization in the loop (ECCV 2020)
- Badger et al. 3D bird reconstruction: a dataset, model, and shape recovery from a single view (ECCV 2020)
- Wang et al. Birds of a feather: Capturing avian shape models from images (CVPR 2021)

* The early important work of Cashman et al. What shape are dolphins? Building 3d morphable models from 2d images (TPAMI 2013) is also not cited.
* Another work that reconstructs dynamic objects from in-the-wild image collections is Wu et al.'s Unsupervised Learning of Probably Symmetric Deformable 3D Objects from Images in the Wild (CVPR 2020 best paper).
* While NeRF and a couple static variants are mentioned, the NeRF field has also produced some works on dynamic reconstruction of general objects from video (not image collections; although see Banmo). These could be added for completion, see e.g. the recent survey Tewari et al. Advances in Neural Rendering (Eurographics 2022).


---- post rebuttal

My rather minor concerns and questions have been addressed well. I increase my rating to accept.

---

> ### Author Response · Authors · 2022-08-02
> **Response to Reviewer mH51**
>
> ---
> **Missing references**
>
> We thank the reviewer for the pointers to additional related works and we will add them to the manuscript.
>
> ---
> **Comparison with BanMo’s semantic loss**
>
> BanMo learns a canonical feature embedding by enforcing the consistency between feature matching and geometric warping, which is made tractable by the dense temporal correspondence (optical flow) between video frames. The semantic loss in BanMo enforces the 2D-3D cycle consistency between 2D coordinates and canonical 3D points. Considering that our image ensemble is sparse and un-correlated, we coarsely initialize and regularize the 3D surface features at the part-level. The proposed semantic loss allows us to first localize the 3D parts then refine the detailed part shapes in the challenging setting.
>
> ---
> **Comparison with additional baselines. Why not compare with CMR, CSM, U-CMR, VMR**
>
> Please see **Comparison with additional baselines** in the General Response-1 above.
>
> ---
> **Evaluations on CUB bird images**
>
> We design LASSIE for articulated shapes and thus its advantage over prior mesh reconstruction methods is best shown in animals like zebras and horses. To demonstrate that LASSIE can also be applied to more compact shapes, we show some quantitative comparisons on CUB bird images in the table below and qualitative results in this [_anonymous link_](https://www.dropbox.com/s/ah0ge6br92y25mp/cub.png?dl=0). Specifically, we select 3 classes (Laysan Albatross, Mallard, and Painted Bunting) in the dataset and filter out the images with truncation or occlusions, resulting in roughly 30 images per class. As shown in  the results, LASSIE produces slightly lower PCK but higher mask IOU compared to UMR.
>
> **Table-T4**: Quantitative comparison on CUB images. We perform optimization and evaluation on each class and report the average metrics.
> | Method     |   PCK\@0.1  |   Mask IOU   |
> | :-----------   |  :-----------:     |  :-----------:     |
> | UMR         |   60.5            |    74.1           |
> | A-CSM      |    45.1          |     76.6           |
> | LASSIE     |    59.8          |     77.3          |
>
> ---
> **Error bars**
>
> In our optimization framework, the random seed initialization does not affect the outputs much since the most sensitive parameters (e.g. camera, bone scaling, pose) are not randomly initialized.
>
> ---
> **Reduced number of images with more diverse poses**
>
> We observe that the limiting factor of few-image optimization is camera viewpoint. That is, with enough diversity of the camera viewpoints in the image ensemble, LASSIE can produce reasonably good results in terms of part discovery and faithfulness to input images. Using images with more diverse poses can produce more realistic and connected part shapes when re-posed or animated.
>
> ---
> **Manual annotations used**
>
> Please see **Manual annotations needed. Is LASSIE self-supervised?** in the General Response-2 above.
>
> ---
> **Clarification of optimization stages**
>
> Please see **Clarification of LASSIE optimization** in the General Response-1 above.
>
> ---
> **Other suggestions**
>
> Thanks for the suggestions. We will incorporate them to improve the paper.

---

> > ### Comment · Reviewer_mH51 · 2022-08-08
> > **Thank you for your response**
> >
> > I appreciate the response, both to my review and the other reviews. At this point, I am satisfied with the answers and don't have any further questions.

---

### Official Review · Reviewer_boNQ · 2022-06-27

**Rating:** 4
**Confidence:** 5
**Soundness:** 2 fair
**Presentation:** 3 good
**Contribution:** 4 excellent

**Summary:**

The paper proposes a new task formulation in the area of non-rigid reconstruction and parts discovery. The model is fit to a small collection of images (25–30) of unrelated instances of a category (such as animal species). It enables discovering 3D pose and shape of objects in all images, along with part segmentation. No supervision on the level of images is required, however the users have to provide the topology and a typical pose of the skeleton, assignment of image parts to image features’ clusters, and also there are specific constraints on the movement of certain joints (such as legs in case of quadrupeds). The model produces better PCK numbers than competing methods (Articulated CSM and 3D Safari) and visually convincing reconstructions on the examples shown.

**Questions:**

Major issues:
* I suggest to allocate more text to position the method w.r.t. UMR [19] more clearly;
* I suggest to evaluate reconstruction quality on datasets with 3D ground truth (see above);
* can you clarify the comparison protocol with the baselines: are they evaluated on the hold-out evaluation set (vs. the proposed method that fits the model to all available images as far as I understood)?
* please address the limitations below.

Other issues:
* In Chamfer loss (3), is the closest point chosen using the distance (4) rather than just geometric distance? If so, what is the motivation of that? I feel like the loss should enforce points to reproject in the correct locations; matching the features should not compensate for that.
* in Experiments, baselines sometimes are trained using a different category; please indicate clearly in the results table where it is the case,


**Limitations:**

The paper lists some limitations, but crucially misses some others:
* Can the method generalise to new images, i.e. can it infer the pose for a new instance without re-fitting the model? The competing methods only need to run the forward pass. That may be important for interactive applications.
* The method uses the same shapes of the individual parts across instances (ll. 172–174), so the resulting meshes differ only in the parts’ rigid transformations. This is an approximations since in reality individuals differ in the bone lengths and deformation of their flesh.
* Another consequence of that is, if I understand correctly, reposing the shape can result in gaps between the parts. In fact, lack of those gaps is guaranteed for training instances only thanks to rendering losses; this means that the invisible surface may not be represented properly,
* the loss L_pose penalises the difference from the rest pose; it is usually difficult to find the weight of such loss so that the method is able to fit extreme poses,
* the need for heuristics like L_ang (5) limits the applicability of the method to new categories where users would need to figure out similar constrints.

**Strengths And Weaknesses:**

Originality:
* I see the method as an extension of UMR [19] to articulated shapes, which are naturally represented as a union of convex’ish primitives, where SCOPS features are upgraded to DINO. Extending to articulated shapes requires overcoming some technical challenges, such as adding regularisations and priors.

Clarity:
* (+) In general, the paper is well written, and the method is mostly clear,
* (−) some of the limitations are either not addressed or not mentioned in the introduction (such as the required category-level supervision), while l. 51 claims that the method is “completely self-supervised”,
* (−) I am confused about the final optimisation algorithm; it runs block-coordinate optimisation (l. 256 and on), but then there is EM-like optimisation of features inside one of the steps; I think the paper would benefit from pseudo-code of the training loop,

Significance:
* (+) getting rid of instance-level supervision is a great step; the method can scale to larger collections easily;
* (+) the required prior information is indeed easier to obtain than e.g. category-specific mesh.

Experiments:
* (+) visual demonstrations look good;
* (−) is the EM-like optimisation important? What would be the result if the original DINO features were used in (4)? (I think it corresponds to 1 iteration of “EM”);
* (−) evaluating using 2D keypoint transfer may be used by other papers but it does not tell much about reconstruction quality; representing the animal’s shape with a flat surface would result in a relatively good PCK while such reconstruction is visually unacceptable. I suggest evaluating on datasets where 3D ground truth is available, such as synthetic datasets or human datasets (such as Human 3.6M) – without expectation of performing better than methods specialising on human reconstruction,
* (+) Applications paragraph/results are nice.

=============

Typos:
* line 80: In contrasted ← in contrast;
* line 146: *k* ← *b*;
* line 211–213: why is it called EM-like algorithm? I would call it block-coordinate descent; for EM, I would expect having an explicit likelihood function depending on latent random variables, and taking expectation over those at E-step,
* line 285: “A-CSM assume high-quality skinned model”: IIRC, it only requires the mesh segmented to parts but no skinning weights, which is a much lighter assumption.
* line 286: “animal bodies which”.

UPD. The other reviewers pointed me towards the supplementary video that I had previously overlooked. It reassured me in the reasonable reconstruction quality, at least on the shown examples. I like the new problem setting and the method. My only remaining (but major) concern is the presentation and positioning of the method and the results: in the current manuscript, the relation to UMR (which is strong, at least on the ideas level) is essentially ignored; Experimental section wrongly claiming conclusive quantitative evaluation of 3D reconstruction; the limitations are not addressed, and the Introduction / abstract massively oversell the method (which it ironically does not even needs). I am changing my rating to borderline to not block the acceptance; I will thus leave it to ACs to decide whether we can accept the paper in this form.

---

> ### Author Response · Authors · 2022-08-02
> **Response on Reviewer boNQ**
>
> ---
> **Comparison with UMR**
>
> Despite that LASSIE and UMR both utilize 2D semantic parts/clusters as supervisory signals, we use the semantic correspondence differently to deal with a fundamentally different problem which UMR cannot be trivially extended to. First, UMR depends on a pre-trained SCOPS model to define part-level correspondence between images and maintain a canonical UV map for semantic rendering, whereas we directly utilize the DINO features to define semantic consistency loss between pixels and 3D surface points. Second, UMR represents the shapes as a single mesh, which is more suitable for compact shapes and can only produce fixed-resolution outputs. Last but not least, UMR cannot perform part-based manipulation or animation given a target pose or motion since it does not model articulations. We also show some quantitative comparisons with UMR in Table-T1 in the General Response-1 above, demonstrating our advantage on articulated shapes like horse bodies. We will add this discussion to the manuscript.
>
> ---
> **Is LASSIE self-supervised?**
>
> Please see **Manual annotations needed. Is LASSIE self-supervised?** in the General Response-2 above.
>
> ---
> **Clarification of optimization algorithm**
>
> Please see **Clarification of LASSIE optimization** in the General Response-1 above.
>
> ---
> **Justification of EM-style feature optimization**
>
> This EM-style semantic and geometry optimization is crucial since it can progressively refine the 3D surface features given better pose and shape fitting, then in return provide more accurate surface-pixel correspondence. Detailed optimization pseudo-code is shown in the General Response-1 above. We justify it by an ablation study of using the original DINO features for pose and shape optimization (one step in EM). As shown in the table below, one step feature optimization leads to significantly worse results since the 3D surface features are only coarsely initialized by the DINO cluster-to-part mapping.
>
> **Table-T3**: Quantitative comparison with/without EM-style feature optimization.
> | Method      |   Horse         |   Zebra         |
> | :-----------    |  :-----------:    |  :-----------:    |
> | One step    | 68.1 / 53.6   | 70.8 / 56.3   |
> | EM-style    | 73.0 / 58.0   | 79.9 / 63.3   |
>
> ---
> **Evaluation using 2D keypoint transfer**
>
> Considering the lack of 3D annotated data for the animal classes of our interest, we follow the common practice in prior works (UMR, CSM, A-CSM, 3D Safari, etc) to evaluate 2D keypoint transfer accuracy. We aim to extend LASSIE to more diverse classes like human bodies in the future work.
>
> ---
> **Chamfer distance in semantic loss**
>
> In Eq. 3, a corresponding pair of 2D pixel and 3D surface point should be close in both the geometric image space and the semantic feature space. By considering both geometric distance and semantic distance, we can pull the 3D points closer to their corresponding pixels by minimizing the semantic loss  (L227-229). Only minimizing the geometric distance can easily result in local minima where the output parts fit the overall silhouettes but not their corresponding semantic clusters.
>
> ---
> **Clarification of evaluation protocol**
>
> The 3D baselines like 3D Safari and A-CSM are trained on large-scale image datasets and evaluated on our image ensemble. LASSIE is optimized and evaluated on all available images in the ensemble (supplemental Table 2). We use the released models of 3D Safari (zebra) and A-CSM (horse, cow, sheep) to evaluate on the closest animal class (L280-282). We will clarify when the baselines are trained on a different class in the evaluation tables.
>
> ---
> **Inference on novel images**
>
> Please see **Inference on novel images** in the General Response-2 above.
>
> ---
> **Part shape constraints**
>
> Please see **Strong pose and shape regularizations** in the General Response-2 above.
>
> ---
> **Pose prior and bone angle losses**
>
> Please see **Strong pose and shape regularizations** in the General Response-2 above.
>
> ---
> **Typo and grammar issues**
>
> Thanks for the corrections. We will revise the paper accordingly.

---

> > ### Comment · Reviewer_boNQ · 2022-08-06
> > **Comparison with UMR**
> >
> > Thank you for your response!
> >
> > Sorry if I was not clear about positioning with respect to UMR. I acknowledge that you solve a different problem, which requires fitting an articulated mesh. However, the big part of the solution of your problem is the "EM-like" optimisation algorithm. Is there a crucial difference between using SCOPS and DINO features in this context? UMR also uses what can be described as an EM-like algorithm, with the averaged SCOPS features acting as latent variables. How specifically does your algorithm differ from it?

---

> > > ### Author Response · Authors · 2022-08-07
> > > **Comparison with UMR**
> > >
> > > **The use of image features and part segmentations in LASSIE and UMR**
> > >
> > > At a high-level, it appears that both UMR and LASSIE start from 2D part segmentations and then lift them onto 3D. A key difference is that LASSIE directly discovers parts in 3D using 2D feature consistency, whereas UMR uses 2D part consistency via canonical UV maps. More specifically, UMR uses 2D parts (not 2D features) and 2D UV maps as common canonical part representations. Instead, LASSIE directly optimizes 3D part features with feature based loss (not 2D parts) and LASSIE does not use UV maps like in UMR. We only use 2D parts for feature initialization, not during optimization. That is, we discover parts directly in 3D with skeleton constraints, without too much reliance on intermediate 2D part discovery. Hence, LASSIE is less sensitive to the issues in 2D part segmentations compared to UMR. Next, we discuss more technical differences in the optimization.
> > >
> > > ---
> > > **EM-style optimization**
> > >
> > > Although LASSIE and UMR both utilize an EM-style strategy for surface feature optimization, there are several differences caused by the problem setting and 2D features we deal with. UMR utilizes a canonical UV map and a pre-defined UV-to-3D mapping to soft-assign mesh surfaces to SCOPS parts. In the E-step, UMR updates the 3D reconstruction network using a part-level probability loss and vertex projection loss through semantic rendering, which are based on the coarse part estimations from SCOPS. In the M-step, the canonical UV maps are updated by averaging across all instances. We observe that learning a high-quality canonical surface map is a suboptimal choice in our framework due to the sparse image setting and multi-surface representation. Instead, we directly exploit the 2D image features (DINO) to define our semantic consistency loss. In our E-step, we assign the average DINO features to the corresponding 3D surfaces, making the 3D surface features more fine-grained compared to UMR. In the M-step, we apply the semantic consistency loss defined between dense 3D points and 2D pixels, unlike the part-level constraints in UMR limited by the number of parts in SCOPS. Note that we only use the 2D feature clusters for feature initialization and thus the semantic consistency loss is not limited by the number of 2D parts.
> > >
> > > ---
> > > **SCOPS v.s. DINO features**
> > >
> > > Even though both DINO and SCOPS are not our contributions, we would like to comment on their differences for the sake of completeness. We find that DINO-ViT features generalize better to novel objects compared to VGG features used in SCOPS part discovery. Moreover, recent work [1] showed that DINO features can discover more consistent and higher resolution parts compared to SCOPS. Our technical contributions in 3D part discovery are orthogonal to the advances in 2D part discovery.
> > >
> > > [1] Amir et al. "Deep vit features as dense visual descriptors." arXiv preprint. 2021.

---

> > > > ### Comment · Reviewer_boNQ · 2022-08-08
> > > > **-**
> > > >
> > > > Thank you; that clarifies the difference well. The paper will benefit from having such description in it.

---

> > > > > ### Author Response · Authors · 2022-08-09
> > > > > **-**
> > > > >
> > > > > Thank you for the feedback. We will clarify the differences to UMR and optimization steps in the paper as well. It would be great if you could take our rebuttal into consideration and update the rating accordingly.

---

> > ### Comment · Reviewer_boNQ · 2022-08-06
> > **Clarification of optimization algorithm**
> >
> > So there are 4 stages, where you gradually extend the range of tuneable parameters, and within each stage, you run this EM-style optimisation? That's much more clear now, thanks.

---

> > > ### Author Response · Authors · 2022-08-07
> > > **Clarification of optimization algorithm**
> > >
> > > **Clarification of optimization algorithm**
> > >
> > > That is correct. We will add the pseudo-code to the supplemental material and clarify it in the manuscript.

---

> > ### Comment · Reviewer_boNQ · 2022-08-06
> > **Evaluation protocol**
> >
> > Thanks once more for clarifications. I find that due to the combination of (1) the method minimising distance between re-projected points, 2) method evaluated on the training set, evaluating PCK is a weak indicator of 3D reconstruction quality. I.e. the method can probably fit a good surface map (which should be easy given DINO features) without reconstructing the correct 3D shape. Hence we can only rely on visuals to evaluate the reconstruction quality.

---

> > > ### Author Response · Authors · 2022-08-07
> > > **Evaluation protocol**
> > >
> > > **Evaluation protocol**
> > >
> > > We acknowledge that 2D keypoint transfer is not a direct 3D evaluation metric, but it is the standard practice in related literature in the absence of 3D annotations. In our experiments, 2D PCK accompanied by visual verification of multi-view results forms a reliable evaluation of 3D reconstruction quality. Although a model can achieve high keypoint transfer accuracy (PCK) by producing accurate canonical surface maps on the source and target images, we argue that the accurate surface maps are not easy to obtain in practice. First, the 2D (DINO) feature maps cannot serve as a canonical surface mapping since several animal parts (e.g. front v.s. back legs, left v.s. right legs, tail hair v.s. head hair) share similar 2D features (see DINO feature clusters in supplemental Figures 10 and 11). We show in the table below that directly using such feature maps will suffer from 2D ambiguity and result in low PCK. On the other hand, defining an accurate 2D-to-3D mapping requires either manual labeling (A-CSM) or fitting on large-scale images (UMR), both of which are not available in our scenario. Moreover, our 3D representation involves combining multiple 3D neural surfaces for articulated shape, imposing more difficulties in defining the surface mapping in 2D. Due to these, our model can only achieve high PCK by an accurate estimation of 3D camera, pose and shapes. Please note that we also include other metrics like mask IOU and part transfer to evaluate the faithfulness to input images and dense correspondence between images. We kindly ask reviewer to take all the quantitative metrics and qualitative comparisons into consideration when evaluating the effectiveness of our method. We hope to extend LASSIE to more object classes with 3D ground-truths in the future work.
> > >
> > > **Table-T5**: Keypoint transfer (PCK\@0.1) evaluation of using DINO features as surface mapping on horse and zebra images. In DINO-nn, we transfer each source keypoint to the target image by finding the pixel with most similar DINO features (nearest neighbor).
> > > | Method     |   Horse          |   Zebra          |
> > > | :-----------   |  :-----------:     |  :-----------:     |
> > > | DINO-nn   | 59.7              | 62.4               |
> > > | LASSIE    | 73.0               | 79.9              |

---

### Official Review · Reviewer_LjEQ · 2022-07-07

**Rating:** 7
**Confidence:** 4
**Soundness:** 3 good
**Presentation:** 4 excellent
**Contribution:** 3 good

**Summary:**

This paper presents an optimization-based methods that reconstructs 3D part based articulated shapes of animals from only a small collection of in-the-wild images (~30 instances) of different instances of an articulated object category, eg, horse, giraffe, elephant, penguin etc. The key idea is leverage self-supervised pretrained ViT features (DINO) to establish semantic correspondences between the images of different instances with various camera viewpoints, pose articulations, texture and environment. By constraining the 3D shapes with a set of pretrained part primitives (ellipse, cylinder, cone) on a pre-defined 3D skeleton (eg, quadrupedal or bipedal), the model is able to reconstruct compelling articulated 3D shapes of different instances, significantly better than existing learning-based models trained on only one specific category (zebra or horse).

**Questions:**

Apart from a few questions that need clarifications listed above, the only additional result I might suggest is to also perform test-time optimization with previous methods for a slightly fairer comparison, maybe in the final version. The settings are different enough though, so this may not really affect the evaluation of the paper.

**Limitations:**

The paper has included a discussion on the limitations of the model as well as illustrative examples in the supplementary material.

**Strengths And Weaknesses:**

## Strengths
### S1 - Compelling results on an interesting and challenging task
- The task of reconstructing articulated 3D animals from only a small collection of in-the-wild images of various articulated instances is highly ill-posed and very challenging. The proposed method seems to work well on a diverse set of examples (9 different animals demonstrated).
- Based on the examples shown in Fig 9 in the supplementary material, the images in each category collection are also very diverse in terms of camera viewpoints, poses, and environment and illumination conditions.

### S2 - Good ideas and careful implementations
- There are two main ideas. One is to leverage self-supervised pretrained image features (DINO ViT) to establish semantic correspondences between different image instances via simple clustering. This is more powerful than previously studied SCOPS which still requires ImageNet pretraining and large training datasets.
- The second idea is to use part-based 3D representation with a pre-defined generic 3D skeleton. This significantly reduces the complexity of this ill-posed task, and yet requires only minimal priors.
- These two ideas are carefully implemented and validated through ablation studies. In particular, the use of an neural surface representation and VAE pretraining of the 3D parts seems quite effective in achieving more regularized shapes.

### S3 - Good writing
- The paper is very well written, with clear motivations, sufficient technical explanations and illustrative visualizations.


## Weaknesses
### W1 - Optimization-based method
- The method optimizes over a small collection of images and does not generalize to new images out of the box. It would also be interesting to extend the idea to a learning-based pipeline that would allow inference on novel images.
- This also makes the comparison against previous learning-based methods unfair, as the results of other methods (I assume) are obtained by one forward pass without any finetuning. It would be slightly fairer to also perform some test time finetuning on the other methods.

### W2 - Heavy regularization
- Due to the ill-posedness of the task, the model relies on a number of heavy constraints, including a simplistic part-based shape representation which also requires pretraining using a set of primitives. This largely limits the expressivity of the model and hence does not lead to fine geoemtric details.
- The need of hand-crafted 3D skeleton is also a major limitation. Although it may seem effortless on one category, it is still quite cumbersome to scale up to all kinds of objects. Moreover, the model only works well on animals that have limited articulated poses, where the skeleton is clearly visible. I can imagine that it would still struggle for animals that highly articulated, like cats (in which case, only a small number of images may not be sufficient).
- There are also a number of specific constraints in the model, eg, minimizing the sideway rotation angles on the leg bones, and a specific optimization procedure that alternates between viewpoints, bones, parts and features.


### (Minor) Clarifications
- Are the part prior latents $\mathbf{e}_i$ randomly initialized and optimized for each part during full training (after VAE pretraining)?
- I am confused by the visualization of the part primitives in Fig 2. Aren't the output of the neural surface networks $\mathcal{F}_i$ supposed to be elongated shapes in the canonical space (without bone orientations)? Fig 2 seems to suggest that the part shapes predicted by the networks are already oriented.
- Line 257: the optimization procedure seems to involve three steps: 1) camera viewpoints, 2) bone transformations, 3) latent part codes and part deformations. Is that correct? What about the 3D part features? When are they updated (the E step)?
- It seems pretty easy for the model to confuse head and tail by looking at only the 4 clusters visualized in the zebra example in Fig 2. Since camera viewpoint is optimized in the first step, will it get confused initially and fall into local minima. And if so, how often does it occur?
- Were the 2D keypoints used in keypoint transfer evaluation manually labelled?


### (Minor) Typos
- Line 146: I believe $k$ should be $b$.
- Line 171: add $t^j \in \mathbb{R}^{b \times 3}$ for completeness. Also, make the notations consistent -- $\mathbf{t}$ vs. $t$.
- Line 286: fix typos.

---

> ### Author Response · Authors · 2022-08-02
> **Response to Reviewer LjEQ**
>
> ---
> **Inference on novel images**
>
> Please see **Inference on novel images** in the General Response-2 above.
>
> ---
> **Unfair comparison with learning-based methods**
>
> We are aware that LASSIE results are not directly comparable to the learning-based methods (L277-230) since we deal with a different problem setting. However, there exists no closer framework and it is non-trivial to extend these baselines to our test-time optimization scenario. Moreover, the test-time optimized results of 3D Safari or A-CSM are also not directly comparable to LASSIE since both methods utilize a shape model fitted on large-scale datasets.
>
> ---
> **Part shape regularizations**
>
> Please see **Strong pose and shape regularizations** in the General Response-2 above.
>
> ---
> **Skeleton-based representation**
>
> While the 3D skeleton can be quite simple and generic (all quadrupeds share the same skeleton in our experiments), it provides a strong and crucial regularization for part transformations. Although cats are not included in our experiments, some qualitative results on tiger images are shown in manuscript Figure 3 and supplemental Figure 11, which are reasonably faithful to the inputs. We observe that LASSIE currently struggles with a) highly articulated parts like elephant trucks (supplemental Figure 12) and b) fluffy animals that appear with more instance variations and ambiguous articulations. In the future work, we hope to solve these issues by discovering 3D skeletons automatically and loosen the rigid part constraints with the help of implicit skeleton representation or temporal correspondence in videos.
>
> ---
> **Bone rotation regularizations**
>
> Please see **Strong pose and shape regularizations** in the General Response-2 above.
>
> ---
> **Clarification of latent part code initialization**
>
> The latent part codes are initialized as zero vectors and optimized for each part during full training (after VAE pretraining).
>
> ---
> **Clarification of part shape visualization in Figure 2**
>
> The output part primitives and deformations are in the canonical space indeed. We show the oriented parts in Figure 2 to indicate the correspondence between individual parts and overall reconstruction. We will revise the figure to avoid confusion.
>
> ---
> **Clarification of optimization process**
>
> Please see **Clarification of LASSIE optimization** in the General Response-1 above.
>
> ---
> **Confusion between head and tail localization**
>
> Since our only image-level supervision is from DINO features, LASSIE can indeed estimate inaccurate camera viewpoints and fall into local minima if the features are noisy and not semantically clustered.  There are 2 failure cases out of the 30 zebra images in our dataset. Note that zebra images have the most ambiguous feature clusters due to their texture, other classes in our experiments do not suffer from this issue (see supplemental Figures 10 and 11).
>
> ---
> **Clarification of keypoint annotations**
>
> The 2D keypoints in our self-collected image ensembles are manually annotated. For Pascal-part images, we automatically find the keypoints by calculating the centers/corners of ground-truth part masks.
>
> ---
> **Typo and grammar issues**
>
> Thanks for the corrections. We will revise the paper accordingly.

---

> > ### Comment · Reviewer_LjEQ · 2022-08-09
> > **Response**
> >
> > I appreciate the authors' detailed response. I like the new experiments on novel image "inference". I am also satisfied with the response to the current limitations and have no further questions. I think this submission present important insight on reconstructing articulated objects from sparse images.

---

### Official Review · Reviewer_rsf7 · 2022-07-10

**Rating:** 7
**Confidence:** 4
**Soundness:** 4 excellent
**Presentation:** 4 excellent
**Contribution:** 4 excellent

**Summary:**

This paper studies self-supervised learning of articulated shapes under a very challenging setting where only a few (~30) in-the-wild images of an animal category are available. This setting is much more challenging than prior art as silhouettes and templates are not used and the number of images used for training is very few. To make this possible, the authors propose to model articulated objects with a 3D skeleton, and each part is modeled with a simple shape. The model is trained via analysis by synthesis, where DINO features are used to provide silhouette and 2D-3D semantic consistency. Experiments on Pascal Part and self-collected web images demonstrate that the proposed method LASSIE achieves considerably better 3D reconstructions and 2D&3D part discovery compared to baselines.

**Questions:**

Please clarify why [39] and [17] are selected instead of the newer ones as mentioned in weaknesses.

**Limitations:**

Yes, the authors adequately addressed the limitations and potential negative societal impact.

**Strengths And Weaknesses:**

Strengths:

1. The proposed method LASSIE is novel and reasonable. The design of modeling each part using simple shape primitives effectively regularizes the solution space and makes the problem more tractable. The 2D-3D semantic consistency loss makes use of the semantically consistent DINO features to help discover 3D parts. The regularization and training procedure are reasonable. I believe these new insights and designs are valuable to the community.

2. Thanks to the proposed method, it is the first time that the self-supervised articulated shape learning task can be addressed under such a challenging setting where only a few (~30) in-the-wild images are available. LASSIE also considerably outperforms previous methods, and has broad applications such as semantic part refinement, pose/texture transfer.

3. The paper is well written.


Weaknesses:

This paper is generally fine. The following weaknesses are not significant.

1. In line 278, it is mentioned that there exists no closer framework other than 3D Safari [39] and A-CSM [17]. However, it seems to me that these are not the state-of-the-art method for self-supervised articulated shape learning. For example, the following papers are later than the two baselines. Please clarify why [39] and [17] are selected instead of these newer ones.
[1] Tulsiani et al, Implicit Mesh Reconstruction from Unannotated Image Collections, NeurIPS2020.
[2] Ye et al, Shelf-Supervised Mesh Prediction in the Wild, CVPR2021.
[3] Li et al, Self-supervised Single-view 3D Reconstruction via Semantic Consistency, ECCV2020.

Typos and grammar issues:
Line 12: We -> we.
Line 146: k denotes -> b denotes.
Line 198: The use of ... -> Through the use of ... or By using ...
Line 223: a image -> an image.
Line 286: an object category is able to produce detailed shapes of animal bodies which -> an object category which is able to produce detailed shapes of animal bodies.

---

> ### Author Response · Authors · 2022-08-02
> **Response to reviewer rsf7**
>
> ---
> **Comparison with additional baselines. Why 3D-Safari and A-CSM are selected instead of newer ones.**
>
> Please see **Comparison with additional baselines** in the General Response-1 above.
>
> ---
> **Typo and grammar issues**
>
> Thanks for the corrections. We will revise the paper accordingly.

---

### Author Response · Authors · 2022-08-02
**General Response-1**

We thank the reviewers for the constructive feedback. For the missing reference suggested, we will add the corresponding discussion and comparisons to the manuscript. We address the common concerns in the general responses (General Response-1, General Response-2) and specific comments in the individual response to each reviewer.

---
**Comparison with additional baselines (R1: rsf7, R4: mH51)**

Due to the lack of prior works on our problem setting (sparse image optimization for articulated animal shapes), we mainly compare LASSIE with learning-based mesh reconstruction methods. Among these methods, we find 3D Safari and A-CSM most comparable to LASSIE since they also model articulation for animal classes of our interest. Most recent mesh reconstruction methods, on the other hand, are weaker baselines since they focus more on compact shapes like birds and cars. In the table below, we show additional quantitative comparisons with CSM, ShSMesh, and UMR on our horse image ensemble (from Pascal-part dataset) in terms of keypoint transfer accuracy (PCK\@0.1) and overall mask IOU. These methods cannot handle articulations and thus perform worse on articulated animals like common quadrupeds. Other related methods suggested by the reviewers either do not release code/model (CMR, IMR, U-CMR) or assume different inputs (VMR).

**Table-T1**: Quantitative comparison with additional baselines on horse images.
| Method     |   PCK\@0.1   |   Mask IOU   |
| :-----------   |  :-----------:     |  :-----------:     |
| CSM          | 50.3             | -                   |
| ShSMesh  | 51.3             | 53.6             |
| UMR          | 55.7             | 58.4             |
| 3D Safari   | 71.8             | 72.2             |
| A-CSM      | 69.3             | 72.5             |
| LASSIE      | 73.0             | 81.9             |

---
**Clarification of LASSIE optimization (R2: LjEQ, R3: boNQ, R4: mH51)**

We provide the pseudo-code below to clarify the optimization process. We will rephrase the alternative optimization to multi-stage optimization, which include 4 main stages: 1) camera, 2) camera and pose, 3) shape, and 4) all parameters. The parameters are optimized using the corresponding losses in each stage. In each iteration, we first update the semantic features of 3D surfaces, then use the updated features to update 3D surfaces, forming an EM-style optimization. The detailed process is shown in the pseudo-code.

---
---
> **Parameters:** resting part rotations $\bar{R}$, bone length scaling {$s_i$}, part rotation {$R^j$}, camera viewpoints {$\pi^j$}, latent part codes {$e_i$}, part deformation MLPs {$\mathcal{F}^\Delta_i$} ($i$: part index, $j$ instance index).

> **Losses:** mask IOU loss $L_{mask}$, semantic consistency loss $L_{sem}$, pose deviation loss $L_{pose}$, part angle prior $L_{ang}$, Laplacian regularization $L_{lap}$, surface normal loss $L_{norm}$.

> **Multi-stage optimization:**
> 1. Optimize {$\pi^j$} using $L_{sem}$ until convergence
> 2. Optimize {$\pi^j$}, $\bar{R}$, {$s_i$}, {$R^j$} using $L_{sem}$, $L_{mask}$, $L_{pose}$, $L_{ang}$ until convergence
> 3. Optimize {$e_i$} and {$\mathcal{F}^\Delta_i$} using  $L_{sem}$, $L_{mask}$, $L_{lap}$, $L_{norm}$ until convergence
> 4. Optimize all parameters using all losses until convergence

> **EM-style semantic and geometric optimization:**
> * Repeat
>   - **E-step:** Update 3D surface features $Q$ by rendering the neural surfaces on each image, finding the nearest pixel for each 3D point, and averaging the corresponding image features.
>   - **M-step:** Optimize neural surfaces using the updated $Q$ in $\mathcal{L}_{sem}$ (Eq. 3). Note that the M-step also involves updating other parameters with different losses depending on the optimization stage.
> * Until end of optimization stage
---
---

---

### Author Response · Authors · 2022-08-02
**General Response-2**

---
**Inference on novel images (R2: LjEQ, R3: boNQ)**

Unlike most prior mesh reconstruction methods, LASSIE deals with a NeRF-like optimization problem with sparse images, and further addresses some additional challenges like unknown cameras, diverse instances, articulations, etc. Therefore, our goal is to produce high-quality and faithful 3D shapes via test-time optimization on few images in-the-wild.

To perform inference on new images without fitting all parameters from scratch, one can fix the shared parameters (resting pose, part features, latent part codes, and part MLPs) optimized on the “training set”, and only optimize the instance-specific camera pose and articulation. In the table below, we show the quantitative comparison on our horse image ensemble. We randomly split the ensemble into training (20) and testing (10)s and report the average results on the testing set. We refer to the original framework as “full optimization”, and the new experiment “partial optimization”. As shown in the results, partial optimization leads to slightly lower PCK and Mask IOU but still performs favorably against prior methods.

**Table-T2**: Full vs. partial optimization on horse test set images.
| Method                      |   PCK\@0.1  |   Mask IOU   |
| :-----------                    |  :-----------:     |  :-----------:    |
| 3D Safari                   | 71.8              | 72.2             |
| A-CSM                      | 69.3              | 72.5             |
| Partial optimization    | 72.0             | 80.1              |
| Full optimization        | 73.0             | 81.9              |

---
**Strong pose and shape regularizations (R2: LjEQ, R3: boNQ)**

We acknowledge the current limitations of our skeleton-based representation and strong regularizations. As the first attempt to address this novel and highly ill-posed problem, we find the proposed pose and shape constraints essential to produce good quality results and avoid unrealistic outputs.

---
+ **Pose and bone angle regularizations.**
Considering our challenging problem setting with sparse images, we find limiting the pose deviation from resting pose crucial to avoid unrealistic poses. Our bone angle loss, on the other hand, is generic to all quadrupeds in our experiments. Similar techniques have been commonly used in prior works (SMALR [1] and 3D Safari for animals, SMPLify [2] for human bodies), either with explicit pose constraints or GMM priors fitted on large datasets. To further alleviate the need of such pre-defined priors, we are working on more general priors like symmetry, gravity, part overlap, bone-axis rotation, etc. For instance, we can find the pitch-yaw-roll axes for each bone based on the initial skeleton and constrain roll-axis rotations to reduce ambiguity during shape optimization.

---
+ **Part shape constraints.**
While the primitive prior MLP produces simple base shapes, individual Part MLPs enable detailed deformation from primitive-like bases. The connectivity between parts is guaranteed by scaling the part surfaces to fit the bone lengths (L142-144). To obtain higher quality shapes, one can perform optimization with high-resolution images and features or instance-specific shape fine-tuning, forming an important future work.

[1] Zuffi et al.. "Lions and tigers and bears: Capturing non-rigid, 3d, articulated shape from images." CVPR. 2018.

[2] Bogo et al. "Keep it SMPL: Automatic estimation of 3D human pose and shape from a single image." ECCV, 2016.

---
**Manual annotations needed. Is LASSIE self-supervised? (R3: boNQ, R4: mH51)**

The proposed LASSIE framework is stated to be “self-supervised” as it does not require any image-level annotations like camera viewpoint, keypoints, or masks. We will clarify this in the manuscript to avoid confusion.

---

### Comment · Area_Chair_u3TK · 2022-08-08
**Any thoughts from the reviewers?**

Hi Reviewers,

The discussion period is closing soon. Please take a look at the responses from the authors. If you have further questions, please ask them now, since the authors will be unable to respond soon. It's substantially more productive, effective, and reasonable to have a quick back-and-forth with authors now than to raise additional questions or concerns post-discussion period that the authors are unable to address.

Thanks,

AC

---

### Meta-Review · Area_Chair_u3TK · 2022-08-22

**Recommendation:** Accept
**Confidence:** Certain

**Metareview:**

This paper had substantial discussion amongst reviewers, and concluded with mixed reviews (7, 7, 7, 4). The positive reviewers (mH51, rsf7,  LjEQ) actively and strongly championed the paper. The remaining concern comes from boNQ, who (in reviewer-to-reviewer discussions) was primarily concerned with making sure that the authors are clear about specifying limitations in method and evaluation. In particular, boNQ was concerned about making sure the following aspects were clearly and prominently discussed in the paper: (a) generalization to new images; (b) assumption of the rest-pose prior; (c) lack of adjusting the shape per-instance; and (d) non-guarantee of gaps between parts. Additionally, boNQ  was concerned about keypoint-based evaluations as a proxy for 3D (although was convinced by the videos in the supplemental).

The AC has examined the paper, reviews, and discussion, and is inclined to agree with the accepting reviewers. The paper will be a strong contribution to the literature and will be of great interest the community. However, the AC notes that many of the reviewers may have similar questions to boNQ. Thus, the AC strongly encourages the authors to address boNQ's concerns in the final version of the paper. With the additional space, the AC believes it will be feasible to specify the limitations more clearly and earlier in the manuscript, and also somehow incorporate additional visualizations demonstrating the effectiveness of the reconstructions. While there is no mechanism for enforcing these changes, they will substantially improve the reception of the paper.

**Award:**

Yes

---

### Decision · Program_Chairs · 2022-09-14

Accept